# A Data-Driven Measure of Relative Uncertainty for Misclassification Detection

**Eduardo Dadalto**[*]
Laboratoire des signaux et systèmes (L2S)
Université Paris-Saclay CNRS CentraleSupélec
Gif-sur-Yvette, France
`eduardo.dadalto@centralesupelec.fr`

**Marco Romanelli**[*]
New York University
New York, NY, USA
`mr6852@nyu.edu`

**Georg Pichler**[*]
Institute of Telecommunications
TU Wien
1040 Vienna, Austria
`georg.pichler@ieee.org`

**Pablo Piantanida**
International Laboratory on Learning Systems (ILLS)
Quebec AI Institute (MILA)
CNRS CentraleSupélec - Université Paris-Saclay
Montreal, Canada
`pablo.piantanida@cnrs.fr`

## Abstract

Misclassification detection is an important problem in machine learning, as it allows for the identification of instances where the model's predictions are unreliable. However, conventional uncertainty measures such as Shannon entropy do not provide an effective way to infer the real uncertainty associated with the model's predictions. In this paper, we introduce a novel data-driven measure of uncertainty relative to an observer for misclassification detection. By learning patterns in the distribution of soft-predictions, our uncertainty measure can identify misclassified samples based on the predicted class probabilities. Interestingly, according to the proposed measure, soft-predictions corresponding to misclassified instances can carry a large amount of uncertainty, even though they may have low Shannon entropy. We demonstrate empirical improvements over multiple image classification tasks, outperforming state-of-the-art misclassification detection methods.

## 1 Introduction

Critical applications, such as autonomous driving and automatic tumor segmentation, have benefited greatly from machine learning algorithms. This motivates the importance of understanding their limitations and urges the need for methods that can detect patterns on which the model uncertainty may lead to dangerous consequences (Amodei et al., 2016). In recent years, considerable efforts have been dedicated to uncovering methods that can deceive deep learning models, causing them to make classification mistakes. While these findings have highlighted the vulnerabilities of deep learning models, it is important to acknowledge that erroneous classifications can also occur naturally. The likelihood of such incorrect classifications is strongly influenced by the characteristics of the data being analyzed and the specific model being used. Even small changes in the distribution of the training and evaluation samples can significantly impact the occurrence of these misclassifications.

A recent thread of research has shown that issues related to misclassifications might be addressed by augmenting the training data for better representation (Zhu et al., 2023a; Zhang et al., 2017; Pinto et al., 2022). However, in order to build misclassification detectors, all these approaches rely on some statistics derived from the soft-prediction output by the model, such as the entropy or related notions, interpreting it as an expression of the model's confidence. We argue that relying on the assumption that the model's output distribution is a good representation of the uncertainty of the model is inadequate. For example, a model may be very confident on a sample that is far from the training distribution and, therefore, it is likely to be misclassified, which undermines the effective use of the Shannon entropy as a measure of the real uncertainty associated with the model's prediction.

---

[*]Equal contribution.

In this work, we propose a data-driven measure of relative uncertainty inspired by Rao (1982) that relies on negative and positive instances to capture meaningful patterns in the distribution of soft-predictions. For example, positive instances can be correctly classified samples for which the uncertainty is expected to be low while negative instances (misclassified samples) are expected to have high uncertainty. Thus, the goal is to yield high and low uncertainty values for negative and positive instances, respectively. **Our measure is "relative", as it is not characterized axiomatically, but only serves the purpose of measuring uncertainty of positive instances relative to negative ones from the point of view of a subjective observer $d$.** We employ relative uncertainty to measure the overall uncertainty of a model, encompassing both aleatoric and epistemic uncertainty components. By learning to minimize the uncertainty in positive instances and to maximize it in negative instances, our metric can effectively capture meaningful information to differentiate between the underlying structure of distributions corresponding to two categories of data. Interestingly, this notion can be expanded to any binary detection tasks in which both positive and negative samples are available.

**Our contributions** are three-fold: **1)** We leverage a novel statistical framework for categorical distributions to devise a learnable measure of relative uncertainty (REL-U) for a model's predictions, which induces large uncertainty for negative instances, even if they may lead to low Shannon entropy (cf. Section 3); **2)** We propose a closed-form solution for training REL-U in the presence of positive and negative instances (cf. Section 4); **3)** We report favorable and consistent results over different models and datasets, considering both natural misclassifications within the same statistical population, and in case of distribution shift, or *mismatch*, between training and testing distributions (cf. Section 5).

## 2 RELATED WORKS

**Misclassification detection** aims to evaluate the reliability of decisions made by classifiers and determine whether they can be trusted or not. A simple baseline relies on the maximum predicted probability (Hendrycks & Gimpel, 2017), but state-of-the-art classifiers have shown to be overconfident in their predictions, even when they fail (Cobb & Looveren, 2022). Liang et al. (2017) proposes applying temperature scaling (Guo et al., 2017) and perturbing the input samples to the direction of the decision boundary to detect misclassifications better. A line of research trains auxiliary parameters to estimate a detection score (Corbière et al., 2019) directly, following the idea of **learning to reject** (Chow, 1970; Geifman & El-Yaniv, 2017). **Exposing the model to outliers or severe augmentations during training** has been explored in previous work (Zhu et al., 2023a) to evaluate if these heuristics are beneficial for this particular task apart from improving robustness to outliers. Granese et al. (2021) proposes a mathematical framework and a simple detection method based on the estimated probability of error. We show that their proposed detection metric is a special case of ours. Zhu et al. (2023b) study the phenomenon that **calibration** methods are most often useless or harmful for failure prediction and provide insights into why. Cen et al. (2023) discusses how training settings such as pre-training or outlier exposure impact misclassification and open-set recognition performance. Related sub-fields are *out-of-distribution detection* (Lee et al., 2018), *open set recognition* (Geng et al., 2021), *novelty detection* (Pimentel et al., 2014), *anomaly detection* (Chalapathy & Chawla, 2019), *adversarial attacks* detection (Akhtar & Mian, 2018), and *predictive uncertainty estimation* via Bayesian Neural Networks estimation (Gal & Ghahramani, 2016; Lakshminarayanan et al., 2016; Mukhoti et al., 2021; Einbinder et al., 2022; Snoek et al., 2019). We refer the reader to Vadera et al. (2020) as a survey in the topic.

A different take on the problem of uncertainty in AI is **conformal learning** Angelopoulos et al. (2021); Romano et al. (2020): in addition to estimating the most likely outcome, a conformal predictor provides a "prediction set" that provably contains the ground truth with high probability. Recently, considerable effort has been invested into quantifying uncertainty by disentangling and estimating two quantities: **epistemic** uncertainty, i.e. the uncertainty that can be decreased by adding new observation to the training set available to a model, and **aleatoric** uncertainty, which is fundamentally present in the data and cannot be reduced by adding training data. In general, these works rely on inducing higher sensitivity at the level of the network's internal representation or on modifications to the training procedure or auxiliary models.Liu et al. (2020) proposes to embed a distance-awareness ability in a given neural network by adding spectral normalization to the weights during training so as to translate the distance in the data manifold into dissimilarity at the hidden representation level. Van Amersfoort et al. (2020) proposes a new loss function and centroid updating scheme to speed up the computation of RBF-based nets.Kotelevskii et al. (2022) proposes an approach

that provides disentanglement of epistemic and aleatoric uncertainty by computing the agreement between a given model and the Bayes classifier based on their kernel estimate and measuring the point-wise Bayes risk. Finally, Mukhoti et al. (2023) utilizes spectral normalization during training and a feature-space density estimator after training to quantify the epistemic uncertainty disentangling it from the aleatoric one.

# 3 A DATA-DRIVEN MEASURE OF UNCERTAINTY

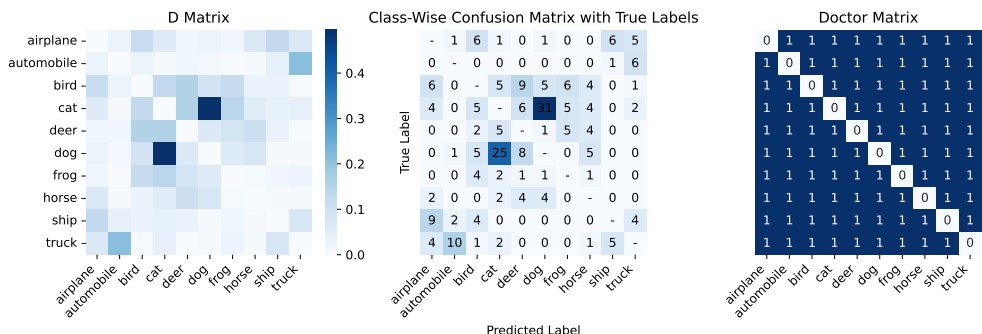

Figure 1: Intuitive example illustrating the advantage of REL-U compared to entropy-based methods: REL-U (left-end side heatmap) captures the real uncertainty (central heatmap) much better than Doctor (Granese et al., 2021); a detailed analysis is provided in Section 5.3.

Before we introduce our method, we start by stating basic definitions and notations. Then, we describe our statistical model and some useful properties of the underlying detection problem.

Let $\mathcal{X} \subseteq \mathbb{R}^d$ be a (possibly continuous) feature space and let $\mathcal{Y} = \{1, \ldots, C\}$ denote the label space related to some task of interest. Moreover, we denote by $P_{XY}$ be the underlying joint probability distribution on $\mathcal{X} \times \mathcal{Y}$. We assume that a machine learning model is trained on some training data, which ultimately yields a model that, given samples $\mathbf{x} \in \mathcal{X}$, outputs a probability mass function (pmf) on $\mathcal{Y}$, which we denote as a vector $\hat{\mathbf{p}}(\mathbf{x})$. This may result from a soft-max output layer, for example. A predictor $f \colon \mathcal{X} \to \mathcal{Y}$ is then constructed, which yields $f(\mathbf{x}) = \arg\max_{y \in \mathcal{Y}} \hat{\mathbf{p}}(\mathbf{x})_y$. We note that we may also interpret $\hat{\mathbf{p}}(\mathbf{x})$ as the probability distribution of $\hat{Y}$ on $\mathcal{Y}$, i.e., given $\mathbf{X} = \mathbf{x}$, $\hat{Y}$ is distributed according to $p_{\hat{Y}|\mathbf{X}}(y|\mathbf{x}) \triangleq \hat{\mathbf{p}}(\mathbf{x})_y$.

In statistics and information theory, many measures of uncertainty were introduced, and some were utilized in machine learning to great effect. Among these are Shannon entropy (Shannon, 1948, Sec. 6), Rényi entropy (Rényi, 1961), $q$-entropy (Tsallis, 1988), as well as several divergence measures, capturing a notion of distance between probability distributions, such as Kullback-Leibler divergence (Kullback & Leibler, 1951), $f$-divergence (Csiszár, 1964), and Rényi divergence (Rényi, 1961). These definitions are well motivated, axiomatically and/or by their use in coding theorems. While some measures of uncertainty offer flexibility by choosing parameters, e.g., $\alpha$ for Rényi $\alpha$-entropy, they are invariant w.r.t. relabeling of the underlying label space. In our case, however, this semantic meaning of specific labels can be important and we do not expect a useful measure of "relative" uncertainty to satisfy this invariance property.

Recall that the quantity $\hat{\mathbf{p}}(\mathbf{x})$ is the soft-prediction output by the model given the input $\mathbf{x}$. The entropy measure of Shannon (Shannon, 1948, Sec. 6)

$$H(\widehat{Y}|\mathbf{x}) \triangleq - \sum_{y \in \mathcal{Y}} \hat{\mathbf{p}}(\mathbf{x})_y \log\left(\hat{\mathbf{p}}(\mathbf{x})_y\right) \tag{1}$$

and the concentration measure of Gini (Gini, 1912)

$$s_{\text{gini}}(\mathbf{x}) \triangleq 1 - \sum_{y \in \mathcal{Y}} \left(\hat{\mathbf{p}}(\mathbf{x})_y\right)^2 \tag{2}$$

have commonly been used to measure the dispersion of a categorical random variable $\widehat{Y}$ given a sample $\mathbf{x}$. It is worth to emphasize that either measure may be used to carry out an analysis of

dispersion for a random variable predicting a discrete value (e.g., a label). This is comparable to the analysis of variance for the prediction of continuous random values.

Regrettably, these measures suffer from two major inconveniences: they are invariant to relabeling of the underlying label space, and, more importantly, they lead to very low values for overconfident predictions, even if they are wrong. These observations make both Shannon entropy and the Gini coefficient unfit for our purpose, i.e., the detection of misclassification instances. Evidently, we need a novel measure of uncertainty that can operate on probability distributions $\hat{\mathbf{p}}(\mathbf{x})$ and that allows us to identify meaningful patterns in the distribution from which uncertainty can be inferred from data. To overcome the aforementioned difficulties, we propose to construct a class of uncertainty measures that is inspired by the measure of diversity investigated in Rao (1982), defined as

$$s_d(\mathbf{x}) \triangleq \mathbb{E}[d(\widehat{Y}, \widehat{Y}')|\mathbf{X} = \mathbf{x}] = \sum_{y \in \mathcal{Y}} \sum_{y' \in \mathcal{Y}} d(y, y')\hat{\mathbf{p}}(\mathbf{x})_y \hat{\mathbf{p}}(\mathbf{x})_{y'}, \qquad (3)$$

where $d \in \mathcal{D}$ is in a class of distance measures and, given $\mathbf{X} = \mathbf{x}$, the random variables $\widehat{Y}, \widehat{Y}' \sim \hat{\mathbf{p}}(\mathbf{x})$ are independently and identically distributed according to $\hat{\mathbf{p}}(\mathbf{x})$. The statistical framework we are introducing here offers great flexibility by allowing for an arbitrary function $d$ that can be learned from data, as opposed to fixing a predetermined distance as in Rao (1982). **In essence, we regard the uncertainty in equation 3 as relative to a given observer $d$, which appears as a parameter in the definition.** To the best of our knowledge, this is a fundamentally novel concept of uncertainty.

## 4 From Uncertainty to Misclassification Detection

We wish to perform misclassification detection based on the statistical properties of soft-predictions of machine learning systems. In essence, the resulting problem requires a binary hypothesis test, which, given a probability distribution over the class labels (the soft-prediction), decides whether a misclassification event likely occurred. We follow the intuition that by examining the soft-prediction of categories corresponding to a given sample, the patterns present in this distribution can provide meaningful information to detect misclassified samples. For example, if a sample is misclassified, this can cause a significant shift in the soft-prediction, even if the classifier is still overconfident. From a broad conceptual standpoint, examining the structure of the population of predicted distributions is very different from the Shannon entropy of a categorical variable. We are primarily interested in the different distributions that we can distinguish from each other by means of positive (correctly classified) and negative (incorrectly classified) instances.

### 4.1 Misclassification Detection Background

We define the indicator of the misclassification event as $E(\mathbf{X}) \triangleq \mathbb{1}[f(\mathbf{X}) \neq Y]$. The occurrence of the "misclassification" event is then characterized by $E = 1$. Misclassification detection is a standard binary classification problem, where $E$ needs to be estimated from $\mathbf{X}$. We will denote the misclassification detector as $g \colon \mathcal{X} \to \{0, 1\}$. The underlying pdf $p_X$ can be expressed as a mixture of two random variables: $\mathbf{X}_+ \sim p_{X|E}(\mathbf{x}|0)$ (positive instances) and $\mathbf{X}_- \sim p_{X|E}(\mathbf{x}|1)$ (negative instances), where $p_{X|E}(\mathbf{x}|1)$ and $p_{X|E}(\mathbf{x}|0)$ represent the pdfs conditioned on the error event and the event of correct classification, respectively.

Let $s \colon \mathcal{X} \to \mathbb{R}$ be the uncertainty measure in (3) that assigns a score $s(\mathbf{x})$ to every sample $\mathbf{x}$ in the input space $\mathcal{X}$. We can derive a misclassification detector $g$ by fixing a threshold $\gamma \in \mathbb{R}$, $g(\mathbf{x}; s, \gamma) = \mathbb{1}[s(\mathbf{x}) \leq \gamma]$, where $g(\mathbf{x}) = 1$ means that the input sample $\mathbf{x}$ is detected as being $E = 1$. In Granese et al. (2021), the authors propose to use the Gini coefficient (2) as a measure of uncertainty, which is equivalent to the Rényi entropy of order two, i.e., $H_2(\widehat{Y}|\mathbf{x}) = -\log \sum_{y \in \mathcal{Y}} (\hat{\mathbf{p}}(\mathbf{x})_y)^2$.

### 4.2 A Data-Driven Measure of Relative Uncertainty for Model's Predictions

We first rewrite $s_d(\mathbf{x})$ (3) in order to make it amenable to learning the metric $d$. By defining the $C \times C$ matrix $D \triangleq (d_{ij})$ using $d_{ij} = d(i, j)$, we have $s_d(\mathbf{x}) = \hat{\mathbf{p}}(\mathbf{x}) D \hat{\mathbf{p}}(\mathbf{x})^\top$. For $s_d(\mathbf{x})$ to yield a good detector $g$, we design a contrastive objective, where we would like $\mathbb{E}[s_d(\mathbf{X}_+)]$, which is the expectation over the positive samples, to be small compared to the expectation over negative samples,

i.e., $\mathbb{E}[s_d(\mathbf{X}_-)]$. This naturally yields the following objective function, where we assume the usual properties of a distance function $d(y, y) = 0$ and $d(y', y) = d(y, y') \geq 0$ for all $y, y' \in \mathcal{Y}$.

**Definition 1.** *Let us introduce our objective function with hyperparameter $\lambda \in [0, 1]$,*

$$\mathcal{L}(D) \triangleq (1 - \lambda) \cdot \mathbb{E}\left[\hat{\mathbf{p}}(\mathbf{X}_+) \, D \, \hat{\mathbf{p}}(\mathbf{X}_+)^\top\right] - \lambda \cdot \mathbb{E}\left[\hat{\mathbf{p}}(\mathbf{X}_-) \, D \, \hat{\mathbf{p}}(\mathbf{X}_-)^\top\right] \tag{4}$$

*and for a fixed $K \in \mathbb{R}^+$, define our optimization problem as follows:*

$$\begin{cases} \text{minimize}_{D \in \mathbb{R}^{C \times C}} \; \mathcal{L}(D) \\ \text{subject to} & d_{ii} = 0, & \forall i \in \mathcal{Y} \\ & d_{ij} \geq 0, & \forall i, j \in \mathcal{Y} \\ & d_{ij} = d_{ji}, & \forall i, j \in \mathcal{Y} \\ & \text{Tr}(DD^\top) \leq K \end{cases} \tag{5}$$

The first constraint in equation 5 states that the elements along the diagonal are zeros, which ensures that the uncertainty measure is zero when the distribution is concentrated at a single point. The second constraint ensures that all elements are non-negative, which is a natural condition, so the measure of uncertainty is non-negative. The natural symmetry between two elements stems from the third constraint, while the last constraint imposes a constant upper-bound on the Frobenius norm of the matrix $D$, guaranteeing that a solution for the underlying optimization problem exists.

**Proposition 1** (Closed form solution). *The constrained optimization problem defined in* (5) *admits a closed form solution $D^* = \frac{1}{Z}(d_{ij}^*)$, where*

$$d_{ij}^* = \begin{cases} \text{ReLU}\left(\lambda \cdot \mathbb{E}\left[\hat{\mathbf{p}}(\mathbf{X}_-)_i^\top \hat{\mathbf{p}}(\mathbf{X}_-)_j\right] - (1 - \lambda) \cdot \mathbb{E}\left[\hat{\mathbf{p}}(\mathbf{X}_+)_i^\top \hat{\mathbf{p}}(\mathbf{X}_+)_j\right]\right) & i \neq j \\ 0 & i = j \end{cases}. \tag{6}$$

*The multiplicative constant $Z$ is chosen such that $D^*$ satisfies the condition $\text{Tr}(D^*(D^*)^\top) = K$.*

The proof is based on a Lagrangian approach and relegated to Appendix A.1. Algorithm 1 in Appendix A.2, summarizes all the main steps for the empirical evaluation, including the data preparation and the computation of the matrix $D^*$.

Note that, apart from the zero diagonal and up to normalization,

$$D^* = \text{ReLU}\left(\lambda \cdot \mathbb{E}\left[\hat{\mathbf{p}}(\mathbf{X}_-)^\top \hat{\mathbf{p}}(\mathbf{X}_-)\right] - (1 - \lambda) \cdot \mathbb{E}\left[\hat{\mathbf{p}}(\mathbf{X}_+)^\top \hat{\mathbf{p}}(\mathbf{X}_+)\right]\right). \tag{7}$$

Finally, we define the Relative Uncertainty (REL-U) score for a given sample $\mathbf{x}$ as

$$s_{\text{REL-U}}(\mathbf{x}) \triangleq \hat{\mathbf{p}}(\mathbf{x}) \, D^* \, \hat{\mathbf{p}}(\mathbf{x})^\top. \tag{8}$$

**Remark.** (2) *is a special case of* (8) *when $d_{ij} = 1$ if $i \neq j$ and $d_{ii} = 0$. Thus, $s_{1-d}(\mathbf{x}) = s_{\text{gini}}(\mathbf{x})$ when choosing $d$ to be the Hamming distance, which was also pointed out in (Rao, 1982, Note 1).*

## 5 EXPERIMENTS AND DISCUSSION

In this section, validate our measure of uncertainty in the context of misclassification detection, considering both the case when the training and test distributions *match* (cf. Section 5.1), and the case in which the two distributions *mismatch* (cf. Section 5.2). Although our method requires additional positive and negative instances, we show that lower amounts are needed (hundreds or few thousands) compared to methods that involve re-training or fine-tuning (hundreds of thousands).

### 5.1 MISCLASSIFICATION DETECTION ON MATCHED DATA

We designed our experiments as follows: for a given model architecture and dataset, we trained the model on the training dataset. We split the test set into two sets: one portion for tuning the detector (held out validation set) and the other for evaluating it. Consequently, we can compute all *hyperparameters* in an unbiased way and cross-validate performance over many splits generated from ten random seeds. For ODIN (Liang et al., 2017) and Doctor (Granese et al., 2021), we found

the best temperature ($T$) and input pre-processing magnitude perturbation ($\epsilon$). For our method, we tuned the best lambda parameter ($\lambda$), $T$, and $\epsilon$. For details on temperature and input pre-processing equations, see Appendix A.6. As of *evaluation metric*, we consider the false positive rate (fraction of misclassifications detected as being correct classifications) when 95% of data is true positive (fraction of correctly classified samples detected as being correct classifications), denoted as FPR at 95% TPR (lower is better). AUROC results are similar among methods (see Figure 6 in the appendix).

Table 1 showcases the misclassification detection performance in terms of FPR at 95% TPR of our method and the strongest baselines (MSP (Hendrycks & Gimpel, 2017), ODIN (Liang et al., 2017), Doctor (Granese et al., 2021)) on different neural network architectures (DenseNet-121 (Huang et al., 2017), ResNet-34 (He et al., 2016)) trained on different datasets (CIFAR-10, CIFAR-100 (Krizhevsky, 2009)) with different learning objectives (Cross-entropy loss, LogitNorm (Wei et al., 2022), MixUp (Zhang et al., 2017), RegMixUp (Pinto et al., 2022), OpenMix (Zhu et al., 2023a)). Please refer to Appendix A.3 for details on the baseline methods. We observe that, on average, our method performs best 11/20 experiments and is equal to the second best in 4/9 out of the remaining experiments. It works consistently better on all the models trained with cross-entropy loss and the models trained with RegMixUp objective, which achieved the best accuracy among them. We observed some negative results when training with logit normalization, but also, the accuracy of the base model decreased. Results for Bayesian methods for uncertainty estimation such as Deep Ensembles (Lakshminarayanan et al., 2016) and MCDropout (Gal & Ghahramani, 2016), as well as results for an MLP directly trained on the tuning data are reported in Table 3 in the Appendix A.6. We report superior detection capabilities for the task at hand.

Table 1: Misclassification detection results across two different architectures trained on CIFAR-10 and CIFAR-100 with five different training losses. We report the average accuracy of these models and the detection performance in terms of average FPR at 95% TPR (lower is better) in percentage with one standard deviation over ten different seeds in parenthesis.

| Model | Training | Accuracy | MSP | ODIN | Doctor | REL-U |
|---|---|---|---|---|---|---|
| DenseNet-121 (CIFAR-10) | CrossEntropy | 94.0 | 32.7 (4.7) | 24.5 (0.7) | 21.5 (0.2) | **18.3** (0.2) |
| | LogitNorm | 92.4 | 39.6 (1.2) | **32.7** (1.0) | 37.4 (0.5) | 37.0 (0.4) |
| | Mixup | 95.1 | 54.1 (13.4) | 38.8 (1.2) | **24.5** (1.9) | 37.6 (0.9) |
| | OpenMix | 94.5 | 57.5 (0.0) | 53.7 (0.2) | 33.6 (0.1) | **31.6** (0.4) |
| | RegMixUp | 95.9 | 41.3 (8.0) | 30.4 (0.4) | 23.3 (0.4) | **22.0** (0.2) |
| DenseNet-121 (CIFAR-100) | CrossEntropy | 73.8 | 45.1 (2.0) | 41.7 (0.4) | **41.5** (0.2) | **41.5** (0.2) |
| | LogitNorm | 73.7 | 66.4 (2.4) | **60.8** (0.2) | 68.2 (0.4) | 68.0 (0.4) |
| | Mixup | 77.5 | 48.7 (2.3) | 41.4 (1.4) | **37.7** (0.6) | **37.7** (0.6) |
| | OpenMix | 72.5 | 52.7 (0.0) | 51.9 (1.3) | 48.1 (0.3) | **45.0** (0.2) |
| | RegMixUp | 78.4 | 49.7 (2.0) | 45.5 (1.1) | 43.3 (0.4) | **40.0** (0.2) |
| ResNet-34 (CIFAR-10) | CrossEntropy | 95.4 | 25.8 (4.8) | 19.4 (1.0) | 14.3 (0.2) | **14.1** (0.1) |
| | LogitNorm | 94.3 | 30.5 (1.6) | **26.0** (0.6) | 31.5 (0.5) | 31.3 (0.6) |
| | Mixup | 96.1 | 60.1 (10.7) | 38.2 (2.0) | 26.8 (0.6) | **19.0** (0.3) |
| | OpenMix | 94.0 | 40.4 (0.0) | 39.5 (1.3) | **28.3** (0.7) | 28.5 (0.2) |
| | RegMixUp | 97.1 | 34.0 (5.2) | 26.7 (0.1) | 21.8 (0.2) | **18.2** (0.2) |
| ResNet-34 (CIFAR-100) | CrossEntropy | 79.0 | 42.9 (2.5) | 38.3 (0.2) | 34.9 (0.5) | **32.7** (0.3) |
| | LogitNorm | 76.7 | 58.3 (1.0) | 55.7 (0.1) | **65.5** (0.2) | **65.4** (0.2) |
| | Mixup | 78.1 | 53.5 (6.3) | 43.5 (1.6) | **37.5** (0.4) | **37.5** (0.3) |
| | OpenMix | 77.2 | 46.0 (0.0) | 43.0 (0.9) | 41.6 (0.3) | **39.0** (0.2) |
| | RegMixUp | 80.8 | 50.5 (2.8) | 45.6 (0.9) | 40.9 (0.8) | **37.7** (0.4) |

Figure 2 displays how the amount of data reserved for the tuning split impacts the performance of the best two detection methods. We demonstrate how our data-driven uncertainty estimation metric generally improves with the amount of data fed to it in the tuning phase, especially on a more challenging setup such as on the CIFAR-100 model.

**Training losses or regularization is independent of detection.** Previous work highlights the independence of training objectives from detection methods, which challenges the meaningfulness of evaluations. In particular, we identify three major limitations in (Zhu et al., 2023a): The evaluation of post-hoc methods, such as Doctor and ODIN, lacks consideration of perturbation and temperature hyperparameters. Despite variations in accuracy and the absence of measures for coverage and risk, different training methods are evaluated collectively. Furthermore, the post-hoc methods are not assessed on these models. The primary flaw in their analysis stems from evaluating different detectors on distinct models, leading to comparisons between (models, detectors) tuples that have different

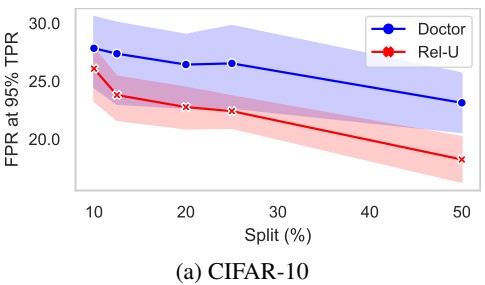 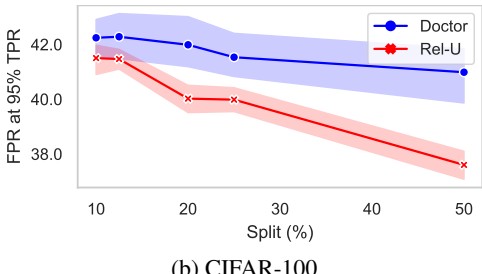

(a) CIFAR-10           (b) CIFAR-100

Figure 2: Impact of the tuning split size on the misclassification performance on a ResNet-34 trained with CE loss for our method and the Doctor baseline. Hyperparameters are set to their default values ($T = 1.0$, $\epsilon = 0.0$, and $\lambda = 0.5$), i.e., only the impact of the validation split size is observed.

misclassification rates. As a result, such an analysis may fail to determine the most performant detection method in real-world scenarios.

**Does calibration improve detection?** There has been growing interest in developing machine learning algorithms that are not only accurate but also well-calibrated, especially in applications where reliable probability estimates are desirable. In this section, we investigate whether models with calibrated probability predictions help improve the detection capabilities of our method or not. Previous work (Zhu et al., 2023b) has shown that calibration does not particularly help or impact misclassification detection on models with similar accuracies, however, they focused only on calibration methods and overlooked detection methods.

To assess this problem in the optics of misclassification detectors, we calibrated the soft-probabilities of the models with a temperature parameter (Guo et al., 2017). Note that this temperature is not necessarily the same value as the detection hyperparameter temperature. This calibration method is simple and effective, achieving performance close to state-of-the-art (Minderer et al., 2021). To measure how calibrated the model is before and after temperature scaling, we measured the expected calibration error (ECE) (Guo et al., 2017) before, with $T = 1$, and after calibration. We obtained the optimal temperature after a cross-validation procedure on the tuning set and measured the detection performance of the detection methods over the calibrated model on the test set. For the detection methods, we use the optimal temperature obtained from calibration, and no input pre-processing is conducted ($\epsilon = 0$), to observe precisely what is the effect of calibration. We set $\lambda = 0.5$.

Table 2 shows the detection performance over the calibrated models. We cannot conclude much from the CIFAR benchmark as the models are already well-calibrated out of the training, with ECE of around 0.03. In general, calibrating the models slightly improved performance on this benchmark. However, for the ImageNet benchmark, we observe that Doctor gained a lot from the calibration, while REL-U remained more or less invariant to calibration on ImageNet, suggesting that the performance of REL-U is robust under the model's calibration.

Table 2: Impact of model probability calibration on misclassification detection methods. The uncalibrated and the calibrated performances are in terms of average FPR at 95% TPR (lower is better) and one standard deviation in parenthesis.

| Architecture | Dataset | $ECE_1$ | $ECE_T$ | Uncal. Doctor | Cal. Doctor | Uncal. REL-U | Cal. REL-U |
|---|---|---|---|---|---|---|---|
| DenseNet-121 | CIFAR-10 | 0.03 | 0.01 | 31.1 (2.4) | 28.2 (3.8) | 32.7 (1.7) | 27.7 (2.1) |
| | CIFAR-100 | 0.03 | 0.01 | 44.4 (1.1) | 45.9 (0.9) | 45.7 (0.9) | 46.6 (0.6) |
| ResNet-34 | CIFAR-10 | 0.03 | 0.01 | 24.3 (0.0) | 23.0 (1.4) | 26.2 (0.0) | 24.2 (0.1) |
| | CIFAR-100 | 0.06 | 0.04 | 40.0 (0.3) | 38.7 (1.0) | 40.6 (0.7) | 38.9 (0.9) |
| ResNet-50 | ImageNet | 0.41 | 0.03 | 76.0 (0.0) | 55.4 (0.7) | 51.7 (0.0) | 53.0 (0.3) |

## 5.2 MISMATCHED DATA

So far, we have evaluated methods for misclassification detection under the assumption that the data available to learn the uncertainty measure and that during testing are drawn from the same distribution.

In this section, we consider cases in which this assumption does not hold true, leading to a mismatch between the generative distributions of the data. Specifically, we investigate two sources of mismatch: *i)* Datasets with different label domains, where the symbol sets and symbols cardinality are different in each dataset; *ii)* Perturbation of the feature space domain generated using popular distortion filters. Understanding how machine learning models and misclassification detectors perform under such conditions can help us gauge and evaluate their robustness.

**Mismatch from different label domains.** We considered pre-trained classifiers on the CIFAR-10 dataset and evaluated their performance in detecting samples in CIFAR-10 and distinguishing them from samples in CIFAR-100, which has a different label domain. Similar experiments have been conducted in Ren et al. (2021); Fort et al. (2021); Zhu et al. (2023a). The test splits were divided into a validation set and an evaluation set, with the validation set consisting of 10%, 20%, 33%, or 50% of the total test split and samples used for training were not reused. For each split, we

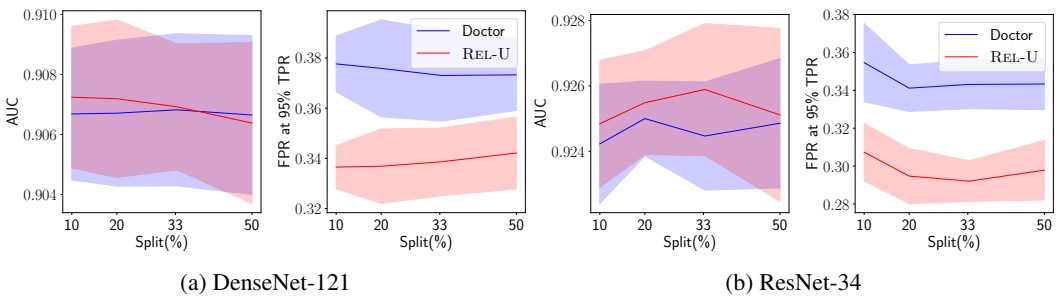

(a) DenseNet-121          (b) ResNet-34

Figure 3: Impact of different validation set sizes (in percentage of test split) for mismatch detection.

combine the number of validation samples from CIFAR-10 with an equal number of samples from CIFAR-100. In order to assess the validity of our results, each split has been randomly selected 10 times, and the results are reported in terms of mean and standard deviation in Figure 3. We observe how our proposed data-driven method performs when samples are provided to accurately describe the two groups. In order to reduce the overlap between the two datasets, and in line with previous work (Fort et al., 2021), we removed the classes in CIFAR-100 that most closely resemble the classes in CIFAR-10. For the detailed list of the removed labels, we refer the reader to Appendix A.7.

**Mismatch from feature space corruption.** We trained a model on the CIFAR-10 dataset and evaluated its ability to detect misclassification on the popular CIFAR-10C corrupted dataset, which contains a version of the classic CIFAR-10 test set perturbed according to 19 different types of corruption and 5 levels of intensities. With this experiment, we aim to investigate if our proposed detector is able to spot misclassifications that arise from input perturbation, based on the sole knowledge of the misclassified patterns within the CIFAR-10 test split.

Consistent with previous experiments, we ensure that no samples from the training split are reused during validation and evaluation. To explore the effect of varying split sizes, we divide the test splits into validation and evaluation sets, with validation sets consisting of 10%, 20%, 33%, or 50% of the total test split. Each split has been produced 10 times with 10 different seeds and the average of the results has been reported in the spider plots in Figure 4. In the case of datasets with perturbed feature spaces, we solely utilize information from the validation samples in CIFAR-10 to detect misclassifications in the perturbed instances of the evaluation datasets, without using corrupted data during validation. We present visual plots that demonstrate the superior performance achieved by our proposed method compared to other methods. Additionally, for the case of perturbed feature spaces, we introduce radar plots, in which each vertex corresponds to a specific perturbation type, and report results for intensity 5. This particular choice of intensity is motivated by the fact that it creates the most relevant divergence between the accuracy of the model on the original test split and the accuracy of the model on the perturbed test split. Indeed the average gap in accuracy between the original test split and the perturbed test split is reported in Table 5 in Appendix A.8.

We observe that our proposed method outperforms Doctor in terms of AUROC and FPR, as demonstrated by the radar plots. As we can see, in the case of CIFAR-10 vs CIFAR-10C, the radar plots (Figure 4) show how the area covered by the AUROC values achieves similar or larger values for the proposed method, indeed confirming that it is able to better detect misclassifications in the

mismatched data. Moreover, the FPR values are lower for the proposed method. For completeness, we report the error bar tables in Tables 6 and 7, Appendix A.8. Additionally, as a particular case of mismatch from feature space corruption, we have considered the task of detecting mismatch between MNIST and SVHN, the results are reported in Figure 7, Appendix A.8.

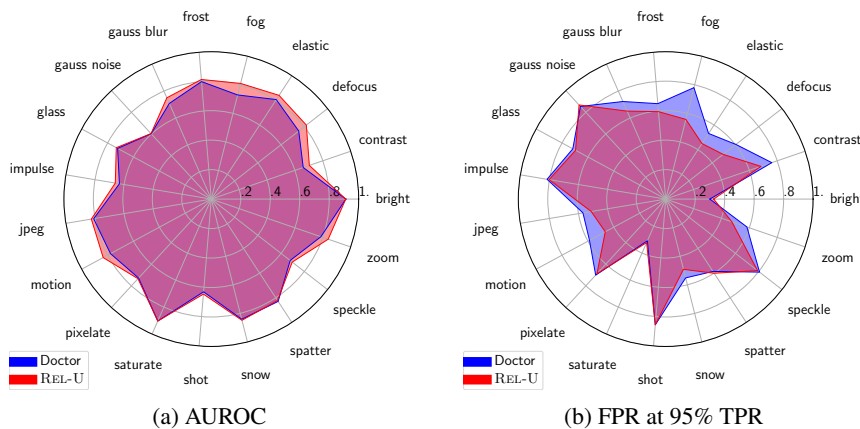

|  |  |
|---|---|
| (a) AUROC | (b) FPR at 95% TPR |

Figure 4: CIFAR-10 vs CIFAR-10C, ResNet-34, using 10% of the test split for validation.

## 5.3 EMPIRICAL INTERPRETATION OF THE RELATIVE UNCERTAINTY MATRIX.

Figure 1 exemplifies the advantage of our method over the entropy-based methods in (1) and (2). In particular, the left-end side heatmap represents the $D$ matrix learned by optimizing (4) on CIFAR-10. Darker shades of blue indicate higher uncertainty, while lighter shades of blue indicate lower uncertainty. The central heatmap is the predictor's class-wise true confusion matrix. The vertical axis represents the true class, while the horizontal axis represents the predicted class. For each combination of two classes $ij$, the corresponding cell reports the count of samples of class $j$ that were predicted as class $i$. The correct matches along the diagonal are dashed for better visualization of the mistakes. The confusion matrix is computed on the same validation set used to compute the $D$ matrix. Crucially, our uncertainty matrix can express different degrees of uncertainty depending on the specific combination of classes at hand. Let us focus for instance on the fact that most of the incorrectly classified dogs are predicted as cats, and vice-versa. The matrix $D$ fully captures this by assigning high uncertainty to the cells at the intersection between these two classes. This is not the case for the entropy-based methods, which cannot capture such a fine-grained uncertainty, and assign the same uncertainty to all the cells, regardless of the specific combination of classes at hand.

## 6 SUMMARY AND CONCLUDING REMARKS

To the best of our knowledge, we are the first to propose REL-U, a **method for uncertainty assessment that departs from the conventional practice of directly measuring uncertainty through the entropy of the output distribution**. REL-U uses a metric that leverages higher uncertainty score for negative data w.r.t. positive data, e.g., incorrectly and correctly classified samples in the context of misclassification detection, and attains favorable results on matched and mismatched data. In addition, our method stands out for its *flexibility and simplicity*, as it relies on a closed form solution to an optimization problem. Extensions to diverse problems present both an exciting and promising avenue for future research.

**Limitations.** We presented machine learning researchers with a fresh methodological outlook and provided machine learning practitioners with a user-friendly tool that promotes safety in real-world scenarios. Some considerations should be put forward, such as the importance of cross-validating the hyperparameters of the detection methods to ensure their robustness on the targeted data and model. As a data-driven measure of uncertainty, to achieve the best performance, it is important to have enough samples at the disposal to learn the metric from as discussed on Section 5.1. As every detection method, our method may be vulnerable to targeted attacks from malicious users.

ACKNOWLEDGEMENTS

This work has been supported by the project PSPC AIDA: 2019-PSPC-09 funded by BPI-France. This work was granted access to the HPC/AI resources of IDRIS under the allocation 2023 - AD011012803R2 made by GENCI.

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

# A   APPENDIX

## A.1   PROOF OF PROPOSITION 1

We have the optimization problem

$$
\begin{cases}
\text{minimize}_{D \in \mathbb{R}^{C \times C}} \ \mathcal{L}(D) \\
\text{subject to} & d_{ii} = 0, \ \forall i \in \{1, \dots, C\}; \\
& d_{ij} - d_{ji} = 0, \ \forall i, j \in \{1, \dots, C\} \\
& \text{Tr}(DD^\top) - K \leq 0 \\
& -d_{ij} \leq 0, \ \forall i, j \in \{1, \dots, C\}
\end{cases}
\tag{9}
$$

in standard form (Boyd & Vandenberghe, 2004, eq. (4.1)) and can thus apply the KKT conditions (Boyd & Vandenberghe, 2004, eq. (5.49)). We find

$$
\nabla \mathcal{L}(D^*) - \sum_{i,j} \xi_{ij}^* \nabla d_{ij}^* + \sum_{i} \mu_i^* \nabla d_{ii}^* + \sum_{ij} \nu_{ij}^* \nabla(d_{ij}^* - d_{ji}^*) + \kappa^* \nabla(\text{Tr}(D^*(D^*)^\top) - K) = 0
\tag{10}
$$

as well as the constraints

$$
d_{ii}^* = 0 \qquad\qquad\qquad d_{ij}^* - d_{ji}^* = 0
\tag{11}
$$

$$
-d_{ij}^* \leq 0 \qquad\qquad\qquad \xi_{ij}^* \geq 0
\tag{12}
$$

$$
\xi_{ij}^* d_{ij} = 0 \qquad\qquad\qquad \kappa^* \geq 0
\tag{13}
$$

$$
\kappa^*(\text{Tr}(D^*(D^*)^\top) - K) = 0
\tag{14}
$$

We have

$$
\nabla \mathcal{L}(D^*) = (1 - \lambda) \cdot \mathbb{E}\left[ \hat{\mathbf{p}}(\mathbf{X}_+)^\top \hat{\mathbf{p}}(\mathbf{X}_+) \right] - \lambda \cdot \mathbb{E}\left[ \hat{\mathbf{p}}(\mathbf{X}_-)^\top \hat{\mathbf{p}}(\mathbf{X}_-) \right]
\tag{15}
$$

$$
\nabla(\text{Tr}(D^*(D^*)^\top) - K) = 2D^*
\tag{16}
$$

and thus[1]

$$
0 = (1 - \lambda) \cdot \mathbb{E}\left[ \hat{\mathbf{p}}(\mathbf{X}_+)^\top \hat{\mathbf{p}}(\mathbf{X}_+) \right] - \lambda \cdot \mathbb{E}\left[ \hat{\mathbf{p}}(\mathbf{X}_-)^\top \hat{\mathbf{p}}(\mathbf{X}_-) \right] - \boldsymbol{\xi}^* + \text{diag}(\boldsymbol{\mu}^*)
$$
$$
+ \boldsymbol{\nu}^* - (\boldsymbol{\nu}^*)^\top + \kappa^* 2D^*
\tag{17}
$$

$$
D^* = \frac{1}{2\kappa^*} \Bigg( -(1 - \lambda) \cdot \mathbb{E}\left[ \hat{\mathbf{p}}(\mathbf{X}_+)^\top \hat{\mathbf{p}}(\mathbf{X}_+) \right] + \lambda \cdot \mathbb{E}\left[ \hat{\mathbf{p}}(\mathbf{X}_-)^\top \hat{\mathbf{p}}(\mathbf{X}_-) \right] + \boldsymbol{\xi}^* - \text{diag}(\boldsymbol{\mu}^*)
$$
$$
- \boldsymbol{\nu}^* + (\boldsymbol{\nu}^*)^\top \Bigg)
\tag{18}
$$

As $\nabla \mathcal{L}(D^*)$ in (15) is already symmetric, we can choose $\boldsymbol{\nu}^* = \mathbf{0}$. We choose[2] $\boldsymbol{\mu}^* = \text{diag}(\nabla \mathcal{L}(D^*))$ to ensures $d_{ii}^* = 0$. The non-negativity constraint can be satisfied by appropriately choosing $\mathbf{0} \leq \boldsymbol{\xi}^* = \text{ReLU}(-\nabla \mathcal{L}(D^*))$. Finally, $\kappa^*$ is chosen such that the constraint $\text{Tr}(D^*(D^*)^\top) = K$ is satisfied. In total, this yields $D^* = \frac{1}{Z} \text{ReLU}(d_{ij}^*)$, where

$$
d_{ij}^* = \begin{cases}
-(1 - \lambda) \cdot \mathbb{E}\left[ \hat{\mathbf{p}}(\mathbf{X}_+)_i^\top \hat{\mathbf{p}}(\mathbf{X}_+)_j \right] + \lambda \cdot \mathbb{E}\left[ \hat{\mathbf{p}}(\mathbf{X}_-)_i^\top \hat{\mathbf{p}}(\mathbf{X}_-)_j \right] & i \neq j \\
0 & i = j
\end{cases}.
\tag{19}
$$

The multiplicative constant $Z = 2\kappa^* > 0$ is chosen such that $D^*$ satisfies the condition $\text{Tr}(D^*(D^*)^\top) = K$.

**Remark.** *A technical problem may occur when $d_{ij}^*$ as defined in (19) is equal to zero for all $i, j \in \{1, 2, \dots, C\}$. In this case, $D^*$ cannot be normalized to satisfy $\text{Tr}(D^*(D^*)^\top) = K$ and the solution to the optimization problem in (9) is the all-zero matrix $D^* = \mathbf{0}$. I.e., no learning is performed in this case. We deal with this problem by falling back to the Gini coefficient (2), where similarly no learning is required.*

*Equivalently, one may also add a small numerical correction $\varepsilon$ to the definition of the $\text{ReLU}$ function, i.e., $\overline{\text{ReLU}}(x) = \max(x, \varepsilon)$. Using this slightly adapted definition when defining $D^* = \frac{1}{Z} \overline{\text{ReLU}}(d_{ij}^*)$ naturally yields the Gini coefficient in this case.*

---

[1]We use $\mathbf{X} = \text{diag}(\mathbf{x})$ for a vector $\mathbf{x}$ to obtain a matrix $\mathbf{X}$ with $\mathbf{x}$ on the diagonal and zero otherwise.

[2]Slightly abusing notation, we also write $\mathbf{x} = \text{diag}(\mathbf{X})$ to obtain the diagonal of the matrix $\mathbf{X}$ as a vector $\mathbf{x}$.

## A.2 ALGORITHM

In this section, we introduce a comprehensive algorithm to clarify the computation of the relative uncertainty matrix $D^*$.

---

**Algorithm 1** Offline relative uncertainty matrix computation.

---

**Require:** $\hat{\mathbf{p}} \colon \mathcal{X} \mapsto \mathbb{R}^C$ trained on a training set with $C$ classes, validation set $\mathcal{D}_m = \{(\mathbf{x}_j, y_j) \underset{\text{i.i.d}}{\sim} p_{XY}\}_{j=1}^m$, and hyperparameter $\lambda \in [0, 1]$

---

$\mathcal{D}_m^+ \leftarrow \varnothing, \ \mathcal{D}_m^- \leftarrow \varnothing$          ▷ Initialize empty positive and negative sets
**for** $(\mathbf{x}, y) \in \mathcal{D}_m$ **do**        ▷ Fill the respective sets with positive or negative samples
    **if** $\arg\max_{y' \in \mathcal{Y}} \hat{\mathbf{p}}(\mathbf{x})_{y'} = y$ **then**
        $\mathcal{D}_m^+ \leftarrow \mathcal{D}_m^+ \cup \{\hat{\mathbf{p}}(\mathbf{x})\}$
    **else**
        $\mathcal{D}_m^- \leftarrow \mathcal{D}_m^- \cup \{\hat{\mathbf{p}}(\mathbf{x})\}$
    **end if**
**end for**
$\boldsymbol{\mu}^+ \leftarrow \frac{1}{|\mathcal{D}_m^+|} \sum_{\hat{\mathbf{p}} \in \mathcal{D}_m^+} \hat{\mathbf{p}}^\top \hat{\mathbf{p}}, \ \boldsymbol{\mu}^- \leftarrow \frac{1}{|\mathcal{D}_m^-|} \sum_{\hat{\mathbf{p}} \in \mathcal{D}_m^-} \hat{\mathbf{p}}^\top \hat{\mathbf{p}}$
$D^* \leftarrow \mathbf{0}_{C \times C}$          ▷ $C$ by $C$ square matrix with zeroed out elements
**for** $i \leftarrow 1, i \leq C, i \leftarrow i + 1$ **do**          ▷ Build $D^*$ according to (6)
    **for** $j \leftarrow 1, j \leq C, j \leftarrow j + 1$ **do**
        **if** $i \neq j$ **then**
            $d_{ij}^* \leftarrow \max\left(\lambda \mu_{ij}^- - (1 - \lambda)\, \mu_{ij}^+, 0\right)$
        **end if**
    **end for**
**end for**
**return** $D^*$

---

At test time, it suffices to compute (8) to obtain the relative uncertainty of the prediction.

## A.3 DETAILS ON BASELINES AND BENCHMARKS

In this section, we provide a comprehensive review of the baselines used on our benchmarks. We state the definitions using our notation introduced in Section 3.

### A.3.1 MSP

The Maximum Softmax Probability (MSP) baseline Hendrycks & Gimpel (2017) proposes to use the confidence of the network as a detection score:

$$s_{\text{MSP}}(\mathbf{x}) = \max_{y \in \mathcal{Y}} \hat{\mathbf{p}}(\mathbf{x})_y \tag{20}$$

### A.3.2 ODIN

Liang et al. (2017) improve upon Hendrycks & Gimpel (2017) by introducing temperature scaling and input pre-processing techniques as described in Appendix A.4, and then compute (20) as the detection score. We tune hyperparameters $T$ and $\epsilon$ on a validation set for each pair of network and training procedure.

### A.3.3 DOCTOR

Granese et al. (2021) propose (2) as the detection score and applies temperature scaling and input pre-processing as described in Appendix A.4. Likewise, we tune hyperparameters $T$ and $\epsilon$ on a validation set for each pair of network and training procedure.

### A.3.4 MLP

We trained an MLP with two hidden layers of 128 units with ReLU activation function and dropout with $p = 0.2$ on top of the hidden representations with a binary cross entropy objective on the

validation set with Adam optimizer and learning rate equal to $10^{-3}$ until convergence. Results on misclassification are presented in Table 3.

### A.3.5 MCDROPOUT

Gal & Ghahramani (2016) propose to approximate Bayesian NNs by performing multiple forward passes with dropout enabled. To compute the confidence score, we averaged the logits and computed the Shannon entropy defined in (1). We set the number of inferences hyperparameter to $k = 10$ and we set the dropout probability to $p = 0.2$. Results on misclassification are presented in Table 3.

### A.3.6 DEEP ENSEMBLES

Lakshminarayanan et al. (2016) propose to approximate Bayesian NNs by averaging the forward pass of multiple models trained on different initializations. We ran experiments with $k = 5$ different random seeds. To compute the confidence score, we averaged logits and computed the MSP response (20). Results on misclassification are presented in Table 3.

### A.3.7 CONFORMAL PREDICTIONS

According to **conformal learning** Angelopoulos & Bates (2021); Angelopoulos et al. (2021); Romano et al. (2020) the presence of uncertainty in predictions is dealt by providing, in addition to estimating the most likely outcome—actionable uncertainty quantification, a "prediction set" that provably "covers" the ground truth with a high probability. This means that the predictor implements an uncertainty set function, i.e., a function that returns a set of labels and guarantees the presence of the right label within the set with a high probability for a given distribution.

### A.3.8 LOGITNORM

Wei et al. (2022) observe that the norm of the logit keeps increasing during training, leading to overconfident predictions. So, they propose Training neural networks with logit normalization to hopefully produce more distinguishable confidence scores between in- and out-of-distribution data. They propose normalizing the logits of the cross entropy loss, resulting in the following loss function:

$$\ell(f(\mathbf{x}), y) = -\log \frac{\exp f_y(\mathbf{x})/(T\|\hat{\mathbf{p}}(\mathbf{x})\|_2)}{\sum_{i=1}^{C} \exp f_i(\mathbf{x})/(T\|\hat{\mathbf{p}}(\mathbf{x})\|_2)}. \tag{21}$$

### A.3.9 MIXUP

Zhang et al. (2017) propose to train a neural network on convex combinations of pairs of examples and their label to minimize the empirical vicinal risk. The mixup data is defined as

$$\tilde{\mathbf{x}} = \lambda \mathbf{x}_i + (1 - \lambda)\mathbf{x}_j \text{ and } \tilde{y} = \lambda y_i + (1 - \lambda)y_j \text{ for } i, j \in \{1, ..., n\}, \tag{22}$$

where $\lambda$ is sampled according to a Beta$(\alpha, \alpha)$ distribution. We used $\alpha = 1.0$ to train the models. Observe a slight abuse of notation here, where $y$ is actually an one-hot encoding of the labels $y = [\mathbb{1}_{y=1}, \ldots, \mathbb{1}_{y=C}]^\top$.

### A.3.10 REGMIXUP

Pinto et al. (2022) use the cross entropy of the mixup data as in (22) with $\lambda$ sampled according to a Beta$(10, 10)$ distribution as a regularizer of the classic cross entropy loss for training a network. The objective is balanced with a hyperparameter $\gamma$, usually set to 0.5.

### A.3.11 OPENMIX

Zhu et al. (2023a) explicitly add an extra class for outlier samples and uses mixup as a regularizer for the cross entropy loss, but mixing between inlier training samples and outlier samples collected from the wild. It yields the objective

$$\mathcal{L} = \mathbb{E}_{\mathcal{D}_{\text{inlier}}} [\ell(f(\mathbf{x}), y)] + \gamma \mathbb{E}_{\mathcal{D}_{\text{outlier}}} [\ell(f(\tilde{\mathbf{x}}), \tilde{y})], \tag{23}$$

where $\gamma \in \mathbb{R}^+$ is a hyperparameter, $\tilde{\mathbf{x}} = \lambda \mathbf{x}_{\text{inlier}} + (1 - \lambda)\mathbf{x}_{\text{outlier}}$, and $\tilde{y} = \lambda y_{\text{inlier}} + (1 - \lambda)(C + 1)$ with a slight abuse of notation. The parameter $\lambda$ is sampled according to a Beta$(10, 10)$ distribution.

## A.4  TEMPERATURE SCALING AND INPUT PRE-PROCESSING

Temperature scaling involves the use of a scalar coefficient $T \in \mathbb{R}^+$ that divides the logits of the network before computing the softmax. This affects the network confidence and the posterior output probability distribution. Larger values of $T$ induce a more flat posterior distribution and smaller values, peakier responses. The final temperature-scaled-softmax function is given by:

$$\sigma(z) = \frac{\exp\left(z/T\right)}{\sum_j \exp\left(z_j/T\right)}.$$

Moreover, the perturbation is applied to the input image in order to increase the network "sensitivity" to the input. In particular, the perturbation is given by:

$$x' = x - \epsilon \times \text{sign}\left[-\nabla_x \log\left(s_{\text{REL-U}}(\mathbf{x})\right]\right,$$

for $\epsilon > 0$. Note that $s_{\text{REL-U}}(\cdot)$ is replaced by the scoring functions of ODIN (20), and Doctor (2) to compute input pre-processing in their respective experiments.

## A.5  ADDITIONAL COMMENTS ON THE ABLATION STUDY FOR HYPER-PARAMETER SELECTION

**Ablation study.** We conducted ablation studies on all relevant parameters: $T$, $\epsilon$, and $\lambda$. It is crucial to emphasize that $T$ is intrinsic to the network architecture and, therefore, must not be considered a hyper-parameter for REL-U. Figure 5 illustrates three ablation studies conducted to analyze and comprehend the effects of different factors on the experimental results. Each subplot represents each hyperparameter ablation study, showcasing the outcomes obtained under specific conditions. **We observe that $\lambda \geq 0.5$, low temperatures, and low noise magnitude achieve better performance.** Overall, the method is shown to be robust to the choices of hyperparameters under reasonable ranges.

Additionally, the introduction of additive noise $\epsilon$ serves the purpose of ensuring a fair comparison with Doctor/ODIN, where the noise was utilized to enhance detection performance. Nevertheless, as indicated by the results in the ablation study illustrated in Figure 5, $\epsilon = 0$ seems to be close to optimal most of the time, thereby positioning REL-U as an effective algorithm that relies only on the soft-probability output, therefore comparable to Granese et al. (2021); Liang et al. (2018) in their version with no perturbation, and Hendrycks & Gimpel (2017). Furthermore, REL-U exhibits a considerable degree of insensitivity to various values of $\lambda$, as evident from Figure 5. This suggests that a potential selection for $\lambda$ could have been $\lambda = N_+/(N_+ + N_-)$, aiming to balance the ratio between the number of positive ($N_+$) and negative ($N_-$) examples. In such a scenario, there are no hyper-parameters at all.

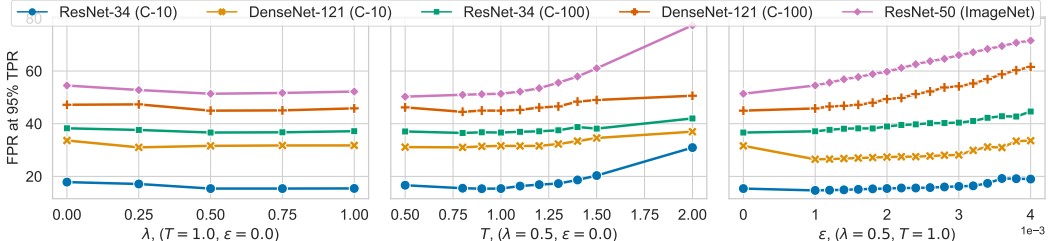

Figure 5: Ablation studies for temperature, lambda, and noise magnitude effects. The x-axis represents the experimental conditions, while the y-axis shows the performance metric.

## A.6  ADDITIONAL RESULTS ON MISCLASSIFICATION DETECTION

**Bayesian methods.**   In this paragraph, we compare our method to additional uncertainty estimation methods, such as Deep Ensembles (Lakshminarayanan et al., 2016), MCDropout (Gal & Ghahramani, 2016), and a MLP directly trained on the validation data used to tune the relative uncertainty matrix. The results are available in Table 3.

Table 3: Misclassification detection results across two different architectures trained on CIFAR-10 and CIFAR-100 with CrossEntropy loss. We report the detection performance in terms of average FPR at 95% TPR (lower is better) in percentage with one standard deviation over ten different seeds in parenthesis.

| Model | Dataset | MCDropout | Deep Ensembles | MLP | REL-U |
|---|---|---|---|---|---|
| DenseNet-121 | CIFAR-10 | 30.3 (3.8) | 25.5 (0.8) | 37.3 (5.8) | **18.3** (0.2) |
| DenseNet-121 | CIFAR-100 | 47.6 (1.2) | 45.9 (0.7) | 78.4 (1.4) | **41.5** (0.2) |
| ResNet-34 | CIFAR-10 | 25.8 (4.9) | 14.8 (1.4) | 33.6 (2.7) | **14.1** (0.1) |
| ResNet-34 | CIFAR-100 | 42.3 (1.0) | 37.4 (1.9) | 63.3 (1.0) | **32.7** (0.3) |

**ROC and Risk-Coverage curves.** We also display the ROC and the risk-coverage curves for our main benchmark on models trained on CIFAR-10 with cross entropy loss. We observe that the performance of REL-U is comparable to other methods in terms of AUROC while outperforming them in high-TPR regions and reducing the risk of classification errors when abstention is desired (coverage) as observed in Figure 6.

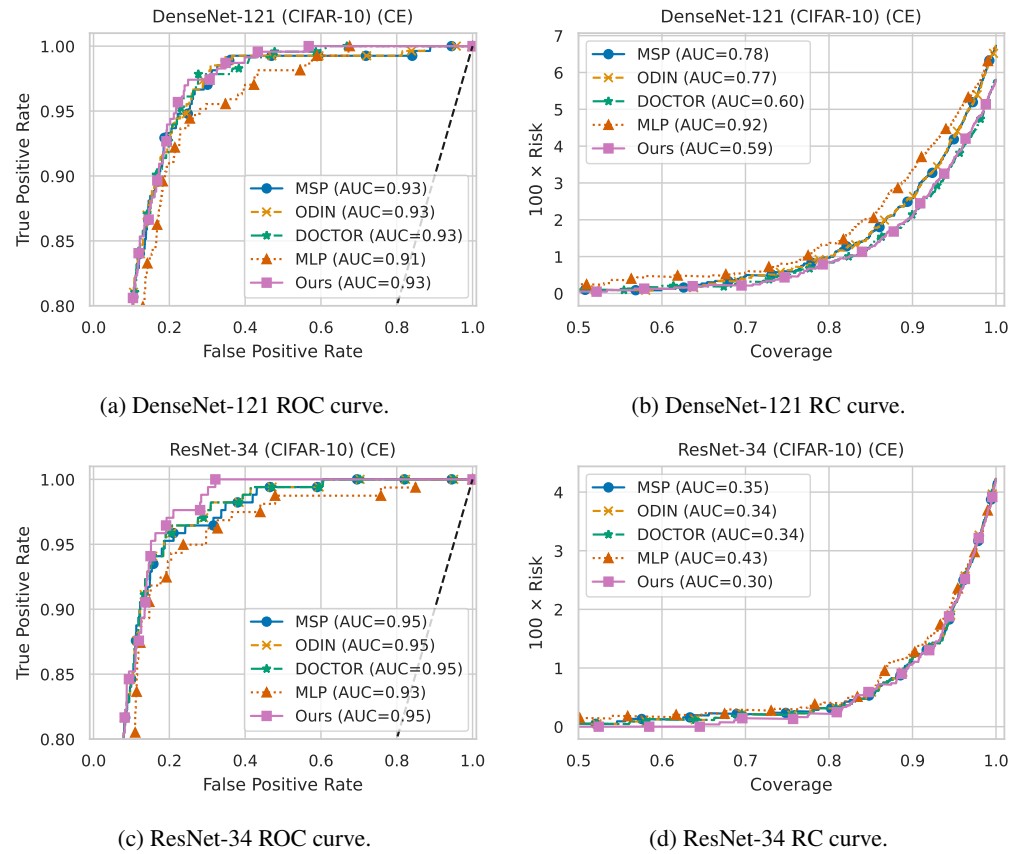

(a) DenseNet-121 ROC curve.

(b) DenseNet-121 RC curve.

(c) ResNet-34 ROC curve.

(d) ResNet-34 RC curve.

Figure 6: Equivalent performance of the detectors in terms of ROC demonstrating lower FPR for our method for high TPR regime. The risk and coverage (RC) curves also looks similar between methods, with a small advantage to our method in terms of AURC.

**Performance of conformal prediction.** We take into account the application of conformal predictors applied to the problem of misclassification. In particular, we consider the excellent work in Angelopoulos & Bates (2021), but most importantly Angelopoulos et al. (2021), which, in turn, builds upon Romano et al. (2020). Conformal predictors, in stark contrast with standard prediction

models, learn a "prediction set function", i.e. they return a set of labels, which should contain the correct value with high probability, for a given data distribution. In particular, Angelopoulos et al. (2021) proposed a revision of Romano et al. (2020), with the main objective of preserving the guarantees of conformal prediction, while, at the same time, minimizing the prediction set cardinality on a sample basis: samples that are "harder" to classify can produce larger sets than samples that are easier to correctly classify. The models are "conformalized" (cf. Angelopoulos et al. (2021)) using the same validation samples, that are also available to the other methods. We reject the decision if the second largest probability within the prediction set exceeds a given threshold, as then, the prediction set would contain more than one label, indicating a possible misclassification event. The experiments are run on 2 models, 2 datasets and 3 training techniques for a total of 12 additional numerical results reported in Table 4. For the model trained with cross entropy in Table 4, the area

Table 4: Misclassification detection results across two different architectures trained on CIFAR-10 and CIFAR-100 with five different training losses. We report the average accuracy of these models and the detection performance in terms of average FPR at 95% TPR (lower is better) in percent with one standard deviation over ten different seeds in parenthesis. The values for the conformalized models are reported in the right-most column.

| Model | Training | Accuracy | MSP | ODIN | Doctor | REL-U | Conf. |
|---|---|---|---|---|---|---|---|
| DenseNet-121 (CIFAR-10) | CrossEntropy | 94.0 | 32.7 (4.7) | 24.5 (0.7) | 21.5 (0.2) | **18.3** (0.2) | 31.6 (3.3) |
| | Mixup | 95.1 | 54.1 (13.4) | 38.8 (1.2) | **24.5** (1.9) | 37.6 (0.9) | 57.6 (6.9) |
| | RegMixUp | 95.9 | 41.3 (8.0) | 30.4 (0.4) | 23.3 (0.4) | **22.0** (0.2) | 30.3 (5.1) |
| DenseNet-121 (CIFAR-100) | CrossEntropy | 73.8 | 45.1 (2.0) | 41.7 (0.4) | **41.5** (0.2) | **41.5** (0.2) | 46.5 (1.3) |
| | Mixup | 77.5 | 48.7 (2.3) | 41.4 (1.4) | **37.7** (0.6) | **37.7** (0.6) | 47.0 (1.3) |
| | RegMixUp | 78.4 | 49.7 (2.0) | 45.5 (1.1) | 43.3 (0.4) | **40.0** (0.2) | 46.0 (1.3) |
| ResNet-34 (CIFAR-10) | CrossEntropy | 95.4 | 25.8 (4.8) | 19.4 (1.0) | 14.3 (0.2) | **14.1** (0.1) | 26.8 (4.6) |
| | Mixup | 96.1 | 60.1 (10.7) | 38.2 (2.0) | 26.8 (0.6) | **19.0** (0.3) | 58.1 (5.6) |
| | RegMixUp | 97.1 | 34.0 (5.2) | 26.7 (0.1) | 21.8 (0.2) | **18.2** (0.2) | 41.9 (7.0) |
| ResNet-34 (CIFAR-100) | CrossEntropy | 79.0 | 42.9 (2.5) | 38.3 (0.2) | 34.9 (0.5) | **32.7** (0.3) | 38.7 (1.5) |
| | Mixup | 78.1 | 53.5 (6.3) | 43.5 (1.6) | **37.5** (0.4) | **37.5** (0.3) | 43.3 (0.9) |
| | RegMixUp | 80.8 | 50.5 (2.8) | 45.6 (0.9) | 40.9 (0.8) | **37.7** (0.4) | 47.7 (1.5) |

under the ROC curve, averaged over 10 seeds, is 0.92 (0.7) for the DenseNet-121 conformalized model on CIFAR-10, and 0.93 (0.7) for the ResNet-34 conformalized model on CIFAR-10, showing comparable results w.r.t. the results in Figures 6a and 6b.

## A.7 MISMATCH FROM DIFFERENT LABEL DOMAINS

In order to reduce the overlap between the label domain of CIFAR-10 and CIFAR-100, in this experimental setup we have ignored the samples corresponding to the following classes in CIFAR-100: bus, camel, cattle, fox, leopard, lion, pickup truck, streetcar, tank, tiger, tractor, train, and wolf.

## A.8 MISMATCH FROM DIFFERENT FEATURE SPACE CORRUPTION

Table 5: We report the gap in accuracy between the original and the corrupted test set for the considered model. The gap is reported and average and standard deviation over the 19 different types of corruptions for corruption intensity equal to 5. The maximum and minimum gap are also reported, with the relative corruption type.

| Architecture | Average gap | Max gap | Min gap |
|---|---|---|---|
| DenseNet121 | $0.36 \pm 0.18$ | 0.66 (Gaussian Blur) | 0.04 (Brightness) |
| ResNet34 | $0.35 \pm 0.20$ | 0.72 (Impulse Noise) | 0.03 (Brightness) |

Table 6: DenseNet-121, error bar table, mismatch from different feature space corruption

| | | Doctor | | REL-U | |
|---|---|---|---|---|---|
| Corruption | Split (%) | AUC | FPR | AUC | FPR |
| Brightness | 10 | $0.90 \pm 0.00$ | $0.31 \pm 0.00$ | $0.90 \pm 0.01$ | $0.35 \pm 0.03$ |
| | 20 | $0.90 \pm 0.00$ | $0.31 \pm 0.00$ | $0.90 \pm 0.00$ | $0.32 \pm 0.01$ |
| | 33 | $0.90 \pm 0.00$ | $0.31 \pm 0.00$ | $0.90 \pm 0.00$ | $0.32 \pm 0.01$ |
| | 50 | $0.90 \pm 0.00$ | $0.31 \pm 0.00$ | $0.90 \pm 0.00$ | $0.32 \pm 0.00$ |
| Contrast | 10 | $0.66 \pm 0.02$ | $0.77 \pm 0.03$ | $0.73 \pm 0.02$ | $0.70 \pm 0.02$ |
| | 20 | $0.66 \pm 0.02$ | $0.77 \pm 0.02$ | $0.73 \pm 0.01$ | $0.69 \pm 0.02$ |
| | 33 | $0.67 \pm 0.01$ | $0.76 \pm 0.01$ | $0.74 \pm 0.01$ | $0.68 \pm 0.01$ |
| | 50 | $0.66 \pm 0.01$ | $0.77 \pm 0.01$ | $0.74 \pm 0.01$ | $0.67 \pm 0.01$ |
| Defocus blur | 10 | $0.70 \pm 0.01$ | $0.75 \pm 0.00$ | $0.72 \pm 0.03$ | $0.71 \pm 0.05$ |
| | 20 | $0.70 \pm 0.01$ | $0.75 \pm 0.00$ | $0.73 \pm 0.01$ | $0.69 \pm 0.01$ |
| | 33 | $0.70 \pm 0.00$ | $0.75 \pm 0.00$ | $0.73 \pm 0.01$ | $0.70 \pm 0.01$ |
| | 50 | $0.70 \pm 0.00$ | $0.75 \pm 0.00$ | $0.73 \pm 0.01$ | $0.71 \pm 0.01$ |
| Elastic transform | 10 | $0.80 \pm 0.01$ | $0.56 \pm 0.00$ | $0.81 \pm 0.01$ | $0.55 \pm 0.02$ |
| | 20 | $0.80 \pm 0.01$ | $0.56 \pm 0.00$ | $0.82 \pm 0.00$ | $0.53 \pm 0.02$ |
| | 33 | $0.80 \pm 0.00$ | $0.56 \pm 0.00$ | $0.82 \pm 0.00$ | $0.53 \pm 0.01$ |
| | 50 | $0.80 \pm 0.00$ | $0.56 \pm 0.00$ | $0.82 \pm 0.00$ | $0.53 \pm 0.01$ |
| Fog | 10 | $0.76 \pm 0.01$ | $0.63 \pm 0.01$ | $0.79 \pm 0.01$ | $0.56 \pm 0.03$ |
| | 20 | $0.76 \pm 0.01$ | $0.63 \pm 0.01$ | $0.79 \pm 0.01$ | $0.55 \pm 0.02$ |
| | 33 | $0.77 \pm 0.00$ | $0.63 \pm 0.01$ | $0.80 \pm 0.00$ | $0.56 \pm 0.02$ |
| | 50 | $0.77 \pm 0.00$ | $0.63 \pm 0.00$ | $0.80 \pm 0.00$ | $0.55 \pm 0.01$ |
| Frost | 10 | $0.78 \pm 0.00$ | $0.62 \pm 0.00$ | $0.79 \pm 0.01$ | $0.61 \pm 0.02$ |
| | 20 | $0.78 \pm 0.00$ | $0.62 \pm 0.00$ | $0.79 \pm 0.01$ | $0.59 \pm 0.02$ |
| | 33 | $0.78 \pm 0.00$ | $0.62 \pm 0.00$ | $0.80 \pm 0.00$ | $0.59 \pm 0.01$ |
| | 50 | $0.78 \pm 0.00$ | $0.62 \pm 0.00$ | $0.80 \pm 0.00$ | $0.59 \pm 0.01$ |
| Gaussian blur | 10 | $0.60 \pm 0.00$ | $0.84 \pm 0.00$ | $0.61 \pm 0.05$ | $0.82 \pm 0.05$ |
| | 20 | $0.60 \pm 0.00$ | $0.84 \pm 0.00$ | $0.63 \pm 0.03$ | $0.82 \pm 0.02$ |
| | 33 | $0.60 \pm 0.00$ | $0.84 \pm 0.00$ | $0.62 \pm 0.02$ | $0.82 \pm 0.01$ |
| | 50 | $0.60 \pm 0.00$ | $0.84 \pm 0.00$ | $0.61 \pm 0.02$ | $0.83 \pm 0.01$ |
| Gaussian noise | 10 | $0.70 \pm 0.00$ | $0.72 \pm 0.00$ | $0.69 \pm 0.02$ | $0.73 \pm 0.02$ |
| | 20 | $0.70 \pm 0.00$ | $0.72 \pm 0.00$ | $0.71 \pm 0.01$ | $0.72 \pm 0.01$ |

| Corruption | Split (%) | | | | |
|---|---|---|---|---|---|
| | 33 | $0.70 \pm 0.00$ | $0.72 \pm 0.00$ | $0.70 \pm 0.01$ | $0.73 \pm 0.01$ |
| | 50 | $0.70 \pm 0.00$ | $0.72 \pm 0.00$ | $0.70 \pm 0.01$ | $0.73 \pm 0.01$ |
| Glass blur | 10 | $0.72 \pm 0.00$ | $0.73 \pm 0.00$ | $0.71 \pm 0.01$ | $0.73 \pm 0.01$ |
| | 20 | $0.72 \pm 0.00$ | $0.73 \pm 0.00$ | $0.72 \pm 0.01$ | $0.72 \pm 0.01$ |
| | 33 | $0.72 \pm 0.00$ | $0.73 \pm 0.00$ | $0.72 \pm 0.01$ | $0.73 \pm 0.00$ |
| | 50 | $0.72 \pm 0.00$ | $0.73 \pm 0.00$ | $0.72 \pm 0.00$ | $0.73 \pm 0.00$ |
| Impulse noise | 10 | $0.62 \pm 0.00$ | $0.85 \pm 0.00$ | $0.61 \pm 0.03$ | $0.84 \pm 0.01$ |
| | 20 | $0.62 \pm 0.00$ | $0.85 \pm 0.00$ | $0.63 \pm 0.02$ | $0.83 \pm 0.01$ |
| | 33 | $0.62 \pm 0.00$ | $0.85 \pm 0.00$ | $0.62 \pm 0.01$ | $0.84 \pm 0.01$ |
| | 50 | $0.62 \pm 0.00$ | $0.85 \pm 0.00$ | $0.62 \pm 0.01$ | $0.84 \pm 0.01$ |
| Jpeg compression | 10 | $0.81 \pm 0.00$ | $0.58 \pm 0.00$ | $0.80 \pm 0.01$ | $0.56 \pm 0.02$ |
| | 20 | $0.81 \pm 0.00$ | $0.58 \pm 0.00$ | $0.80 \pm 0.00$ | $0.55 \pm 0.01$ |
| | 33 | $0.81 \pm 0.00$ | $0.58 \pm 0.00$ | $0.81 \pm 0.00$ | $0.55 \pm 0.01$ |
| | 50 | $0.81 \pm 0.00$ | $0.58 \pm 0.00$ | $0.81 \pm 0.00$ | $0.55 \pm 0.01$ |
| Motion blur | 10 | $0.78 \pm 0.01$ | $0.63 \pm 0.00$ | $0.81 \pm 0.01$ | $0.56 \pm 0.02$ |
| | 20 | $0.78 \pm 0.01$ | $0.63 \pm 0.00$ | $0.82 \pm 0.01$ | $0.53 \pm 0.02$ |
| | 33 | $0.78 \pm 0.00$ | $0.63 \pm 0.00$ | $0.82 \pm 0.00$ | $0.54 \pm 0.02$ |
| | 50 | $0.78 \pm 0.00$ | $0.63 \pm 0.00$ | $0.82 \pm 0.00$ | $0.54 \pm 0.01$ |
| Pixelate | 10 | $0.68 \pm 0.00$ | $0.82 \pm 0.00$ | $0.68 \pm 0.03$ | $0.80 \pm 0.01$ |
| | 20 | $0.68 \pm 0.00$ | $0.82 \pm 0.00$ | $0.67 \pm 0.03$ | $0.81 \pm 0.01$ |
| | 33 | $0.68 \pm 0.00$ | $0.82 \pm 0.00$ | $0.66 \pm 0.02$ | $0.81 \pm 0.01$ |
| | 50 | $0.68 \pm 0.00$ | $0.82 \pm 0.00$ | $0.67 \pm 0.02$ | $0.81 \pm 0.01$ |
| Saturate | 10 | $0.89 \pm 0.00$ | $0.37 \pm 0.01$ | $0.88 \pm 0.01$ | $0.39 \pm 0.03$ |
| | 20 | $0.89 \pm 0.00$ | $0.37 \pm 0.01$ | $0.88 \pm 0.00$ | $0.36 \pm 0.01$ |
| | 33 | $0.89 \pm 0.00$ | $0.37 \pm 0.00$ | $0.88 \pm 0.00$ | $0.37 \pm 0.01$ |
| | 50 | $0.89 \pm 0.00$ | $0.37 \pm 0.00$ | $0.88 \pm 0.00$ | $0.36 \pm 0.01$ |
| Shot noise | 10 | $0.71 \pm 0.00$ | $0.72 \pm 0.00$ | $0.72 \pm 0.02$ | $0.72 \pm 0.02$ |
| | 20 | $0.71 \pm 0.00$ | $0.72 \pm 0.00$ | $0.73 \pm 0.01$ | $0.70 \pm 0.02$ |
| | 33 | $0.71 \pm 0.00$ | $0.72 \pm 0.00$ | $0.73 \pm 0.01$ | $0.70 \pm 0.01$ |
| | 50 | $0.71 \pm 0.00$ | $0.72 \pm 0.00$ | $0.73 \pm 0.01$ | $0.71 \pm 0.01$ |
| Snow | 10 | $0.81 \pm 0.00$ | $0.60 \pm 0.00$ | $0.81 \pm 0.01$ | $0.57 \pm 0.01$ |
| | 20 | $0.81 \pm 0.00$ | $0.60 \pm 0.00$ | $0.81 \pm 0.01$ | $0.57 \pm 0.02$ |
| | 33 | $0.81 \pm 0.00$ | $0.60 \pm 0.00$ | $0.81 \pm 0.00$ | $0.57 \pm 0.01$ |
| | 50 | $0.81 \pm 0.00$ | $0.60 \pm 0.00$ | $0.81 \pm 0.00$ | $0.57 \pm 0.00$ |
| Spatter | 10 | $0.78 \pm 0.00$ | $0.80 \pm 0.00$ | $0.77 \pm 0.02$ | $0.80 \pm 0.04$ |
| | 20 | $0.78 \pm 0.00$ | $0.80 \pm 0.00$ | $0.77 \pm 0.01$ | $0.79 \pm 0.03$ |
| | 33 | $0.78 \pm 0.00$ | $0.80 \pm 0.00$ | $0.77 \pm 0.01$ | $0.80 \pm 0.02$ |
| | 50 | $0.78 \pm 0.00$ | $0.80 \pm 0.00$ | $0.77 \pm 0.00$ | $0.80 \pm 0.02$ |
| Speckle noise | 10 | $0.73 \pm 0.00$ | $0.68 \pm 0.00$ | $0.74 \pm 0.02$ | $0.67 \pm 0.03$ |
| | 20 | $0.73 \pm 0.00$ | $0.68 \pm 0.00$ | $0.75 \pm 0.01$ | $0.65 \pm 0.02$ |
| | 33 | $0.73 \pm 0.00$ | $0.68 \pm 0.00$ | $0.75 \pm 0.01$ | $0.65 \pm 0.01$ |
| | 50 | $0.73 \pm 0.00$ | $0.68 \pm 0.00$ | $0.75 \pm 0.01$ | $0.66 \pm 0.01$ |
| Zoom blur | 10 | $0.73 \pm 0.01$ | $0.72 \pm 0.01$ | $0.76 \pm 0.01$ | $0.67 \pm 0.04$ |
| | 20 | $0.73 \pm 0.01$ | $0.71 \pm 0.00$ | $0.76 \pm 0.01$ | $0.65 \pm 0.02$ |
| | 33 | $0.73 \pm 0.00$ | $0.72 \pm 0.00$ | $0.77 \pm 0.01$ | $0.66 \pm 0.02$ |
| | 50 | $0.73 \pm 0.00$ | $0.72 \pm 0.00$ | $0.77 \pm 0.01$ | $0.67 \pm 0.01$ |

Table 7: ResNet-34, error bar table, mismatch from different feature space corruption

| | | Doctor | | REL-U | |
|---|---|---|---|---|---|
| Corruption | Split (%) | AUC | FPR | AUC | FPR |

| | | | | | |
|---|---|---|---|---|---|
| Brightness | 10 | $0.91 \pm 0.00$ | $0.30 \pm 0.02$ | $0.91 \pm 0.01$ | $0.33 \pm 0.06$ |
| | 20 | $0.91 \pm 0.00$ | $0.30 \pm 0.01$ | $0.92 \pm 0.00$ | $0.30 \pm 0.02$ |
| | 33 | $0.91 \pm 0.00$ | $0.30 \pm 0.01$ | $0.92 \pm 0.00$ | $0.30 \pm 0.01$ |
| | 50 | $0.92 \pm 0.00$ | $0.30 \pm 0.01$ | $0.92 \pm 0.00$ | $0.31 \pm 0.01$ |
| Contrast | 10 | $0.66 \pm 0.03$ | $0.76 \pm 0.03$ | $0.70 \pm 0.02$ | $0.68 \pm 0.03$ |
| | 20 | $0.66 \pm 0.02$ | $0.76 \pm 0.03$ | $0.71 \pm 0.01$ | $0.67 \pm 0.02$ |
| | 33 | $0.66 \pm 0.02$ | $0.75 \pm 0.02$ | $0.72 \pm 0.01$ | $0.66 \pm 0.02$ |
| | 50 | $0.66 \pm 0.01$ | $0.75 \pm 0.01$ | $0.72 \pm 0.01$ | $0.66 \pm 0.01$ |
| Defocus blur | 10 | $0.75 \pm 0.02$ | $0.60 \pm 0.01$ | $0.82 \pm 0.01$ | $0.49 \pm 0.01$ |
| | 20 | $0.75 \pm 0.01$ | $0.60 \pm 0.01$ | $0.82 \pm 0.01$ | $0.49 \pm 0.01$ |
| | 33 | $0.76 \pm 0.01$ | $0.60 \pm 0.00$ | $0.82 \pm 0.00$ | $0.50 \pm 0.01$ |
| | 50 | $0.76 \pm 0.01$ | $0.60 \pm 0.00$ | $0.82 \pm 0.00$ | $0.50 \pm 0.01$ |
| Elastic transform | 10 | $0.81 \pm 0.02$ | $0.53 \pm 0.01$ | $0.84 \pm 0.01$ | $0.45 \pm 0.01$ |
| | 20 | $0.81 \pm 0.01$ | $0.52 \pm 0.01$ | $0.85 \pm 0.00$ | $0.44 \pm 0.01$ |
| | 33 | $0.81 \pm 0.01$ | $0.52 \pm 0.00$ | $0.85 \pm 0.00$ | $0.44 \pm 0.01$ |
| | 50 | $0.81 \pm 0.01$ | $0.52 \pm 0.00$ | $0.85 \pm 0.00$ | $0.44 \pm 0.00$ |
| Fog | 10 | $0.73 \pm 0.02$ | $0.78 \pm 0.05$ | $0.81 \pm 0.01$ | $0.56 \pm 0.02$ |
| | 20 | $0.73 \pm 0.01$ | $0.77 \pm 0.03$ | $0.81 \pm 0.01$ | $0.57 \pm 0.03$ |
| | 33 | $0.74 \pm 0.01$ | $0.77 \pm 0.03$ | $0.81 \pm 0.01$ | $0.59 \pm 0.02$ |
| | 50 | $0.74 \pm 0.01$ | $0.77 \pm 0.02$ | $0.82 \pm 0.00$ | $0.59 \pm 0.03$ |
| Frost | 10 | $0.80 \pm 0.00$ | $0.65 \pm 0.02$ | $0.81 \pm 0.01$ | $0.60 \pm 0.05$ |
| | 20 | $0.80 \pm 0.00$ | $0.65 \pm 0.01$ | $0.82 \pm 0.00$ | $0.59 \pm 0.02$ |
| | 33 | $0.80 \pm 0.00$ | $0.65 \pm 0.01$ | $0.82 \pm 0.00$ | $0.59 \pm 0.01$ |
| | 50 | $0.80 \pm 0.00$ | $0.65 \pm 0.01$ | $0.82 \pm 0.00$ | $0.58 \pm 0.01$ |
| Gaussian blur | 10 | $0.71 \pm 0.01$ | $0.72 \pm 0.00$ | $0.75 \pm 0.01$ | $0.65 \pm 0.01$ |
| | 20 | $0.71 \pm 0.00$ | $0.72 \pm 0.00$ | $0.75 \pm 0.01$ | $0.66 \pm 0.01$ |
| | 33 | $0.71 \pm 0.00$ | $0.72 \pm 0.00$ | $0.75 \pm 0.00$ | $0.66 \pm 0.01$ |
| | 50 | $0.71 \pm 0.00$ | $0.72 \pm 0.00$ | $0.75 \pm 0.00$ | $0.67 \pm 0.01$ |
| Gaussian noise | 10 | $0.60 \pm 0.00$ | $0.85 \pm 0.01$ | $0.60 \pm 0.03$ | $0.87 \pm 0.02$ |
| | 20 | $0.60 \pm 0.00$ | $0.85 \pm 0.01$ | $0.61 \pm 0.01$ | $0.87 \pm 0.01$ |
| | 33 | $0.60 \pm 0.00$ | $0.85 \pm 0.00$ | $0.61 \pm 0.01$ | $0.87 \pm 0.01$ |
| | 50 | $0.60 \pm 0.00$ | $0.85 \pm 0.00$ | $0.61 \pm 0.01$ | $0.87 \pm 0.00$ |
| Glass blur | 10 | $0.72 \pm 0.00$ | $0.72 \pm 0.00$ | $0.73 \pm 0.01$ | $0.70 \pm 0.03$ |
| | 20 | $0.72 \pm 0.00$ | $0.72 \pm 0.00$ | $0.74 \pm 0.01$ | $0.69 \pm 0.01$ |
| | 33 | $0.72 \pm 0.00$ | $0.72 \pm 0.00$ | $0.74 \pm 0.00$ | $0.70 \pm 0.01$ |
| | 50 | $0.72 \pm 0.00$ | $0.71 \pm 0.00$ | $0.74 \pm 0.00$ | $0.69 \pm 0.00$ |
| Impulse noise | 10 | $0.63 \pm 0.00$ | $0.82 \pm 0.00$ | $0.66 \pm 0.02$ | $0.80 \pm 0.03$ |
| | 20 | $0.63 \pm 0.00$ | $0.82 \pm 0.00$ | $0.66 \pm 0.01$ | $0.80 \pm 0.01$ |
| | 33 | $0.63 \pm 0.00$ | $0.82 \pm 0.00$ | $0.66 \pm 0.01$ | $0.80 \pm 0.01$ |
| | 50 | $0.63 \pm 0.00$ | $0.82 \pm 0.00$ | $0.67 \pm 0.01$ | $0.80 \pm 0.00$ |
| Jpeg compression | 10 | $0.81 \pm 0.01$ | $0.57 \pm 0.02$ | $0.82 \pm 0.01$ | $0.51 \pm 0.03$ |
| | 20 | $0.81 \pm 0.01$ | $0.56 \pm 0.01$ | $0.83 \pm 0.00$ | $0.50 \pm 0.01$ |
| | 33 | $0.81 \pm 0.00$ | $0.57 \pm 0.01$ | $0.83 \pm 0.00$ | $0.51 \pm 0.01$ |
| | 50 | $0.81 \pm 0.00$ | $0.57 \pm 0.00$ | $0.83 \pm 0.00$ | $0.51 \pm 0.01$ |
| Motion blur | 10 | $0.78 \pm 0.01$ | $0.59 \pm 0.02$ | $0.83 \pm 0.01$ | $0.47 \pm 0.01$ |
| | 20 | $0.78 \pm 0.01$ | $0.58 \pm 0.01$ | $0.84 \pm 0.01$ | $0.47 \pm 0.01$ |
| | 33 | $0.78 \pm 0.01$ | $0.58 \pm 0.01$ | $0.84 \pm 0.00$ | $0.48 \pm 0.01$ |
| | 50 | $0.78 \pm 0.00$ | $0.57 \pm 0.00$ | $0.84 \pm 0.00$ | $0.48 \pm 0.01$ |
| Pixelate | 10 | $0.73 \pm 0.00$ | $0.70 \pm 0.01$ | $0.73 \pm 0.02$ | $0.69 \pm 0.04$ |
| | 20 | $0.73 \pm 0.00$ | $0.70 \pm 0.01$ | $0.74 \pm 0.02$ | $0.69 \pm 0.03$ |
| | 33 | $0.73 \pm 0.00$ | $0.70 \pm 0.01$ | $0.74 \pm 0.01$ | $0.69 \pm 0.02$ |
| | 50 | $0.73 \pm 0.00$ | $0.70 \pm 0.00$ | $0.74 \pm 0.01$ | $0.68 \pm 0.01$ |
| Saturate | 10 | $0.90 \pm 0.00$ | $0.31 \pm 0.01$ | $0.90 \pm 0.01$ | $0.32 \pm 0.08$ |

| | | | | | |
|---|---|---|---|---|---|
| | 20 | $0.90 \pm 0.00$ | $0.31 \pm 0.00$ | $0.91 \pm 0.00$ | $0.30 \pm 0.01$ |
| | 33 | $0.90 \pm 0.00$ | $0.31 \pm 0.00$ | $0.91 \pm 0.00$ | $0.30 \pm 0.01$ |
| | 50 | $0.90 \pm 0.00$ | $0.31 \pm 0.00$ | $0.91 \pm 0.00$ | $0.29 \pm 0.01$ |
| Shot noise | 10 | $0.63 \pm 0.00$ | $0.86 \pm 0.01$ | $0.65 \pm 0.03$ | $0.86 \pm 0.04$ |
| | 20 | $0.63 \pm 0.00$ | $0.85 \pm 0.01$ | $0.65 \pm 0.01$ | $0.86 \pm 0.01$ |
| | 33 | $0.63 \pm 0.00$ | $0.86 \pm 0.00$ | $0.65 \pm 0.01$ | $0.86 \pm 0.02$ |
| | 50 | $0.63 \pm 0.00$ | $0.86 \pm 0.00$ | $0.65 \pm 0.01$ | $0.86 \pm 0.00$ |
| Snow | 10 | $0.84 \pm 0.00$ | $0.55 \pm 0.03$ | $0.85 \pm 0.01$ | $0.49 \pm 0.03$ |
| | 20 | $0.84 \pm 0.00$ | $0.55 \pm 0.02$ | $0.85 \pm 0.00$ | $0.48 \pm 0.02$ |
| | 33 | $0.84 \pm 0.00$ | $0.55 \pm 0.01$ | $0.85 \pm 0.00$ | $0.48 \pm 0.02$ |
| | 50 | $0.84 \pm 0.00$ | $0.56 \pm 0.01$ | $0.85 \pm 0.00$ | $0.48 \pm 0.01$ |
| Spatter | 10 | $0.83 \pm 0.00$ | $0.59 \pm 0.02$ | $0.82 \pm 0.01$ | $0.60 \pm 0.06$ |
| | 20 | $0.83 \pm 0.00$ | $0.58 \pm 0.01$ | $0.83 \pm 0.01$ | $0.58 \pm 0.04$ |
| | 33 | $0.83 \pm 0.00$ | $0.59 \pm 0.01$ | $0.83 \pm 0.01$ | $0.58 \pm 0.02$ |
| | 50 | $0.83 \pm 0.00$ | $0.59 \pm 0.00$ | $0.83 \pm 0.00$ | $0.58 \pm 0.01$ |
| Speckle noise | 10 | $0.68 \pm 0.00$ | $0.81 \pm 0.01$ | $0.70 \pm 0.03$ | $0.79 \pm 0.06$ |
| | 20 | $0.68 \pm 0.00$ | $0.81 \pm 0.01$ | $0.70 \pm 0.01$ | $0.78 \pm 0.03$ |
| | 33 | $0.68 \pm 0.00$ | $0.81 \pm 0.00$ | $0.70 \pm 0.01$ | $0.79 \pm 0.02$ |
| | 50 | $0.68 \pm 0.00$ | $0.81 \pm 0.00$ | $0.70 \pm 0.01$ | $0.79 \pm 0.01$ |
| Zoom blur | 10 | $0.79 \pm 0.01$ | $0.58 \pm 0.01$ | $0.84 \pm 0.01$ | $0.47 \pm 0.02$ |
| | 20 | $0.79 \pm 0.01$ | $0.58 \pm 0.00$ | $0.84 \pm 0.00$ | $0.48 \pm 0.01$ |
| | 33 | $0.79 \pm 0.01$ | $0.58 \pm 0.00$ | $0.84 \pm 0.00$ | $0.49 \pm 0.01$ |
| | 50 | $0.79 \pm 0.00$ | $0.58 \pm 0.00$ | $0.84 \pm 0.00$ | $0.49 \pm 0.01$ |

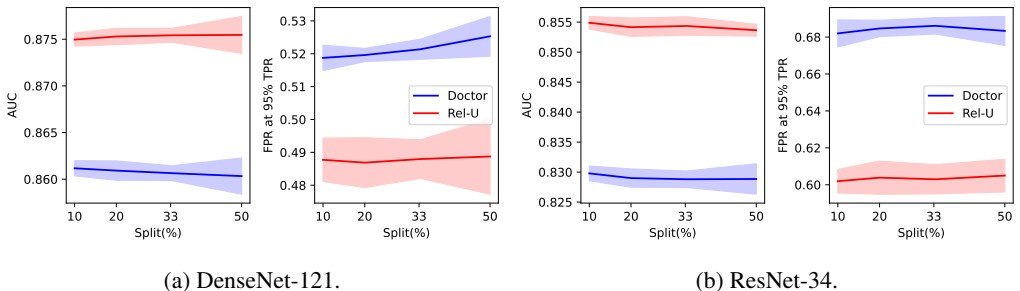

(a) DenseNet-121.         (b) ResNet-34.

Figure 7: SVHN versus MNIST mismatch analysis.

