# OpenReview forum: "A Data-Driven Measure of Relative Uncertainty for Misclassification Detection"
_ICLR.cc/2024/Conference — ICLR 2024 poster_

### Official Review · Reviewer_tQ2Y · 2023-10-22

**Soundness:** 2 fair
**Presentation:** 2 fair
**Contribution:** 2 fair
**Rating:** 6
**Confidence:** 2

**Summary:**

The paper proposes a new measure to determine predictions’ uncertainty based on a distance between different labels that is found using additional data. Such distance can be found solving an optimization problem in closed form and the authors experimentally show the methods introduced obtain good results

**Strengths:**

The addressed topic is very relevant since the probabilistic assessment provided by supervised learning algorithms are often unreliable. The uncertainty metric introduced can be of interest and the fact that the distance between labels can be found in closed form is also valuable

**Weaknesses:**

In order to assess the paper contribution, the authors need to compare their methods with stronger techniques to assess predictive uncertainty, such as conformal prediction. For instance the authors could compare their method with a conformal predicition algorithm that instead of using data to obtain the uncertainty metric s_d, uses such data as calibration data to obtain prediction intervals. Such comparison is also important in that the assessment of prediction uncertainty in terms of prediction sets seems to be more useful in practice than that in terms of the metric s_d(x)

**Questions:**

Can you articulate the meaning of the metric in (3) in more precise terms? refer the distance d as an observer is a bit strange, what does the data-based distance d(y,y') describe?

It is not correct to refer to p_x,y as “probability density function” since Y has a finite range of values

---

> ### Author Response · Authors · 2023-11-16
> **Official answers to Reviewer tQ2Y**
>
> We thank reviewer tQ2Y for their helpful feedback. In the following, we address each of their concerns individually.
> ## Weaknesses:
> - We thank the reviewer for bringing this to our attention. In order to address this point, we have added a dedicated section on conformal prediction in the related work section, including a short explanation in section A.3.7 in the appendix, and most importantly a new set of experiments in section A.6 (Performance of conformal prediction). We considered 2 modesl, 2 datasets and 3 possible training techniques for a total of 12 additional numerical results. The content related to these changes is reported in blue in the revised version of the paper.
> ## Questions:
>
> - The weights within the $D$ matrix can be interpreted as a measure of dissimilarity between the soft-prediction values for all pairs of classes $(y,y’)$ (higher values indicate higher uncertainty). However, it's crucial to note that the notion of similarity or dissimilarity is contextual and relative to the positive and negative examples within the validation data. Therefore, it is "relative" in the sense that it describes the observer in relation to this specific dataset.
> - For every y, the function $p_{X|Y}$ is a probability density function, while $p_Y$ is a probability mass function. In order to avoid complicated expressions when referring to $p_{X,Y}$, we slightly abuse terminology and also call it a probability density function.
>
> We believe that all the concerns and questions were adequately addressed, and kindly ask the reviewer to consider raising their scores accordingly. We are happy to provide any further clarification where needed.

---

> > ### Comment · Reviewer_tQ2Y · 2023-11-17
> >
> > I thank the authors for adding the comparison with conformal prediction methods. Such comparison can be useful to position the paper so I will increase my score to 6.
> >
> > The meaning of the distance (or matrix D) is still not totally clear to me. Regarding the reference of p_x,y it would be better if the authors use a more correct term such as probability measure or probability distribution.

---

> > > ### Author Response · Authors · 2023-11-20
> > > **Re: Official Comment by Reviewer tQ2Y**
> > >
> > > Thank you for carefully reviewing our answers and the revised manuscript.
> > >
> > > We agree with the suggested terminology for the symbol  $p_{x,y}$. We updated the manuscript accordingly, referring to it as a probability distribution.
> > >
> > > According to [Rao, 1982], D must be chosen to reflect some intrinsic dissimilarity between instances relevant to a particular task at hand. In practice, in our paper, the positive definite matrix D implicitly defines an inner product, which in general measures the similarity of vectors. We use data to optimize this inner product for the task of detecting misclassifications.
> > >
> > > **References**
> > >
> > > [Rao, 1982] Rao, C. R.
> > > Diversity and dissimilarity coefficients: a unified approach
> > > *Theoretical population biology, Elsevier*, 1982, 21, 24-43

---

### Official Review · Reviewer_YaSF · 2023-10-28

**Soundness:** 3 good
**Presentation:** 2 fair
**Contribution:** 2 fair
**Rating:** 6
**Confidence:** 3

**Summary:**

This paper proposes to evaluate how classification model predictions tend to be wrong without knowing the groundtruth label. It proposes to calculate a distance matrix of different classes, and use it to calculate a measure for the task. Empirical evaluations using different model structures and data are conducted to compare the proposed method with existing entropy measures.

**Strengths:**

- The paper considers a practically important problem.
- It proposes a principled method and shows how to calculate with clear formulas.
- The proposed idea has a certain degree of originality.
- Thorough experiments are conducted and the source code is provided for reproduction.

**Weaknesses:**

- The paper is overall not easy to follow. For example, the related work section comes at a random place. Also, I encourage authors to conduct wider and more structured literature reviews, to cover related topics such as deep model calibration and classification with rejection, and present in a more structured way.
  - Please also give reivew on the measure of diversity investigated in Rao (1982), and how the proposed method is inspired. This accounts for the core contribution for the paper.
- Terminology is not unifed. For example, "soft-prediction", "the posterior distribution", "the predicted distrbution" is used, which seem to mean one same thing.
- Some words are used without clear definition or in a confusing way.
  - "feature" is used in cases where it seems to mean a data instance, such as an image.
  - lambda parameter for the proposed method is mentioned in experiments, but there is no definition.
- Presentation overall needs to be polished. See questions below.
  - At first, the presentation of "positive" and "negative" is confusing. I was even thinking about binary classification for a second.
  - In contribution 2, "closed-form solution for training REL-U" is confusing. In my opinion, if it is closed-form, usually is not considered as a traning process.

**Questions:**

- How to prepare positive and negative data for calculating the distance matrix at the first place?
- What does match and mismatch mean in experiments?
- Why use auc instead of the metric mentioned in Granese et al. 2021?

---

> ### Author Response · Authors · 2023-11-16
> **Official answers to Reviewer YaSF**
>
> We thank reviewer YaSF for their helpful feedback. In the following, we address each of their concerns individually.
> ## Weaknesses:
> - We thank the reviewer for the observations. We moved the related works section up and placed it after the introduction. We also structured it better, by highlighting important related domains, adding additional comments and references pointed out in the review process.
>
>    The measure investigated in Rao (1982) is introduced in Eq. (3). We improved the phrasing, to highlight this fact.
> - We invested significant efforts in standardizing our terminology. We refer the reviewer to the updated manuscript.
> - We thank the reviewer for pointing out these concerns. We made the necessary changes, updating the terminology and refer the reviewer to the current version of the manuscript.
> The variable $\lambda$ is a real number in $[0,1]$. It is introduced in Definition 1, and the related eq. 4.
> - We thank the reviewer for their suggestions. We added an example of positive and negative samples right after these terms are introduced and rephrased some text to eliminate any possible ambiguity between an iterative learning algorithm and the optimization of the objective function with Lagrangian multipliers.
> ## Questions:
> - We refer the reviewer to the newly introduced Algorithm 1 in the Appendix A.2 where we detailed how to prepare negative and positive data for the computation of the $D^*$ matrix.
> - We investigate two sources of mismatch:
>
>    1. Datasets with different label domains, which translates to experiments of CIFAR-10 vs CIFAR-100 and vice versa or CIFAR-10 vs SVHN for instance. \
>    2. Perturbation of the feature space domain generated using popular distortion filters, which translates to running experiments on CIFAR-C datasets [A].
>
> - We thank the reviewer for pointing out this typo. Actually, we meant AUROC, not just  AUC. The evaluation metrics are indeed the same as in Granese et al. 2021.
>
> We believe that all the concerns and questions were adequately addressed, and kindly ask the reviewer to consider raising their scores accordingly. We are happy to provide any further clarification where needed.
>
> **References**
>
> [A] Hendrycks, Dan, and Thomas Dietterich. "Benchmarking Neural Network Robustness to Common Corruptions and Perturbations." ICLR 2019. /abs/1903.12261.

---

> > ### Comment · Reviewer_YaSF · 2023-11-20
> >
> > Thank you for your detailed response. I have read the updated manuscript and other comments.
> > The presentation of the manuscript is significantly improved due to authors effort. Thus I have raised my score.
> >
> > I also have some concerns remained.
> > - What is the motivation for introducing lambda?
> > - Experiments are also conducted on extrame lambda values such as 0 and 1. Did this cause some unstablity on solving Eq.4?
> > - The proposed method seems to rely on a validation set for preparing positive and negative samples to solve Eq.4. Although it is usually not difficult to prepare in realistic settings, do baseline methods compared in experiments also rely on such a validation set?

---

> > > ### Author Response · Authors · 2023-11-20
> > > **Re: Official Comment by Reviewer YaSF**
> > >
> > > Thank you for carefully reviewing our answers and the revised manuscript.
> > > Regarding the remaining concerns, let us address these individually below:
> > >
> > > - The introduction of $\lambda$ allows us to weigh positive and negative samples differently. This tradeoff is especially useful in the common scenario where there is significant data imbalance. Also refer to the answers to Question 1 of reviewer Pj5j and to Weakness 2 of 9edT.
> > > - We did not encounter any issues with numerical instabilities for extreme values of $\lambda$ using our closed-form solution (6).
> > > - Yes, also the baseline methods rely on additional samples, at least for determining the acceptance/rejection threshold. Some methods, like ODIN and Doctor, also require additional validation samples for tuning parameters $\epsilon$ and $T$.
> > >
> > > The additional samples provided to every method were, of course, the same in every case to ensure a fair evaluation.

---

> > > > ### Comment · Reviewer_YaSF · 2023-11-21
> > > >
> > > > Thank you for clear and prompt response. My concerns have been resolved. The manuscript as well as the authors response now form a clear presentation of the method, and I would have assigned a higher score if shown at the first place. Thus I raised my score to 6.

---

### Official Review · Reviewer_Pj5j · 2023-10-29

**Soundness:** 3 good
**Presentation:** 3 good
**Contribution:** 3 good
**Rating:** 8
**Confidence:** 4

**Summary:**

In the paper, the authors propose a novel approach to learn a measure of "relative" uncertainty based on data.
By utilising both positive (correctly classified) and negative (misclassified) instances, they construct a CxC matrix, which is needed for the proposed method.
This is an interesting approach, as typically to estimate uncertainty we use some ready-to-use formulas (like entropy or maxprob). And to the best of my knowledge, this is the first idea when misclassifications are directly used to derive a measure of uncertainty.

**Strengths:**

1. The paper is well written and I find it easy to follow.
2. The concept of learning a "relative" uncertainty measure from correctly and incorrectly classified data is innovative and novel.
3. The authors have provided a comprehensive experimental evaluation.
4. Authors provided code.

**Weaknesses:**

There are no major drawbacks that I see. But I think the following could improve the paper even further:
1. A more detailed discussion / description of the baselines, such as Doctor, would be beneficial. The same applies to different objectives like Mixup, RegMixUp etc.
2. I think it also should be mentioned that the paper is actually not the first one introducing a data-driven measure of uncertainty. There is a whole batch of works (see for example [1, 2, 3, 4]) that use data to derive estimates of epistemic or/and aleatoric uncertainties. I think it could help readers to correctly place the current paper in the right position in the broader research landscape.
3. A discussion on how "relative uncertainty" compares to well-known epistemic and aleatoric uncertainties would be insightful.

---

[1] Kotelevskii N. et al. Nonparametric Uncertainty Quantification for Single Deterministic Neural Network //Advances in Neural Information Processing Systems.

[2] Van Amersfoort J. et al. Uncertainty estimation using a single deep deterministic neural network //International conference on machine learning.

[3] Liu J. et al. Simple and principled uncertainty estimation with deterministic deep learning via distance awareness //Advances in Neural Information Processing Systems.

[4] Mukhoti J. et al. Deep deterministic uncertainty: A simple baseline //arXiv preprint arXiv:2102.11582.

**Questions:**

Here I have not only questions, but also some comments.

1. Choice of lambda: can we choose it based on the data? Like $\lambda \sim \frac{1}{\text{Npos.examples}}$
2. In page 9: "no class labels required", but it is not really true. You need class labels to make positive/negative pairs. Saying this, I mean that we cannot use the approach in semi-supervised scenario, when objects without labels are available. We still need to know labels.
3. The confusion matrix in Figure 1, as I understood, was computed using the validation data. Shouldn't it be more fair to build it on test set?
4. In scenarios where there's a very good classifier and a high-quality dataset (like MNIST or CIFAR10, with low aleatoric uncertainty), there might be a significant imbalance between positive and negative samples. This could lead to the situation that the trained uncertainty measure is unreliable. How should one address this situation?

---

> ### Author Response · Authors · 2023-11-16
> **Official answers to Reviewer Pj5j**
>
> We thank reviewer Pj5j for their helpful feedback. In the following, we address each of their concerns individually.
>
>
> ## Weaknesses:
> 1. We added discussion on the baselines and objective functions. Please see Section A.3 in the appendix.
> 2. We thank the reviewer for pointing out the references [1-4]. They were added and properly cited in the discussion on Related Works.
> 3.  We employ relative uncertainty to measure the overall uncertainty of a model, encompassing a mixture of both aleatoric and epistemic uncertainty components. On a principled basis, our method, like any approach solely utilizing the model output and validation samples, does not anticipate capturing specifically either aleatoric or epistemic uncertainty. This limitation arises from inherent constraints within the model architecture. We have added a clarification in the revised version of the paper.
> ## Questions:
> 1. While Rel-U does not appear sensitive to changes in λ as shown in Figure 3, a reasonable choice would be $\lambda = N_+ / (N_+ + N_-)$ in order to balance the effect of positive and negative examples. We added a discussion in Appendix A.5.
> 2. We thank the reviewer for bringing this useful comment to our attention. We meant that strictly speaking, eq. 4 only needs positive and negative instances, which are task-specific. Of course in our case, the labels are implicitly needed to produce negative and positive instances. Thus, we removed the sentence from the paper.
> 3. In order to obtain the matrix D we utilize the validation set. Figure 1 illustrates that D can approximate relatively well the confusion matrix for the same samples that were used for computing it.
> 4. This might be addressed in part by tuning the $\lambda$ parameters as mentioned above. However, if the imbalance is too severe (e.g., near perfect classification), it is questionable whether misclassification detection is feasible (or even suitable) in such a setting. This would certainly require further analysis and could be a possible direction for future research.
>
> We believe that all the concerns and questions were adequately addressed, and we are happy to provide any further clarification where needed.
>
> **References:**
>
> [1] Kotelevskii N. et al. Nonparametric Uncertainty Quantification for Single Deterministic Neural Network //Advances in Neural Information Processing Systems.
>
> [2] Van Amersfoort J. et al. Uncertainty estimation using a single deep deterministic neural network //International conference on machine learning.
>
> [3] Liu J. et al. Simple and principled uncertainty estimation with deterministic deep learning via distance awareness //Advances in Neural Information Processing Systems.
>
> [4] Mukhoti J. et al. Deep deterministic uncertainty: A simple baseline //arXiv preprint arXiv:2102.11582.

---

> > ### Comment · Reviewer_Pj5j · 2023-11-22
> >
> > I would like to thank the authors for their answers.
> >
> > They well addressed my questions.
> >
> > I will keep my original score.

---

### Official Review · Reviewer_9edT · 2023-11-03

**Soundness:** 3 good
**Presentation:** 2 fair
**Contribution:** 2 fair
**Rating:** 6
**Confidence:** 3

**Summary:**

In this paper, the authors present a technique to learn their proposed uncertainty measure for misclassification detection. The learned uncertainty measure has a closed form solution which makes it easy to implement. The authors present a wide range of empirical results comparing against baselines trained with different loss functions, and looking at the ablation of various hyperparameters of their approach.

**Strengths:**

- The paper is well motivated and looks at the useful problem of identifying potential errors that models can make to avoid downstream problems.

- The paper provides a theoretically sound approach to learning the uncertainty metric.

- The ablation study provided by the authors is important in understanding the approach better given the # of hyperparameters in the approach.

**Weaknesses:**

- I think the paper can improve on its presentation. Most importantly, the authors should provide a clear description of how the learn their metric function $s_d$ because that's the crux of their paper.

- While the ablation results provided by the authors are useful, this approach still requires a lot of knobs to be tuned, especially if someone is interested in applying it to a completely different application domain.

**Questions:**

- I think the paper would benefit from having an algorithm block just to clarify the exact learning procedure. For instance, information like the detector is tuned on a held-out set which the model wasn't trained out should be highlighted much clearly in the paper.

- A high-level question here is: how is this different from some logit/feature transformation on held-out set? There has been some work in the literature (although around OOD detection) that looks at using latent representations from deep neural network backbone and training models that use them (see for e.g. [1]). I see that you have taken a slightly different approach here that looks at using closed form solution, but the idea seems very similar. Can you describe how your paper differentiates from this high-level approach of taking features from a deep net backbone and then applying some transformation to it to generate an uncertainty metric?

- Building upon the prev point, would it make a lot of difference if you were to use a simple MLP architecture rather than the closed form solution you have right now to transform the soft outputs into an uncertainty measure that's being fine-tuned on held-out tuning set? The MLP can also be trained on held-out tuning set.

- How do you think your approach compares with Bayesian neural networks which also introduce other forms of uncertainty such as model uncertainty? (see [2] for e.g.)



References

[1] Dinari, O. &amp; Freifeld, O.. (2022). Variational- and metric-based deep latent space for out-of-distribution detection. <i>Proceedings of the Thirty-Eighth Conference on Uncertainty in Artificial Intelligence</i>, in <i>Proceedings of Machine Learning Research</i> 180:569-578 Available from https://proceedings.mlr.press/v180/dinari22a.html.

[2] Vadera, M., Li, J., Cobb, A., Jalaian, B., Abdelzaher, T., & Marlin, B. (2022). URSABench: A system for comprehensive benchmarking of Bayesian deep neural network models and inference methods. Proceedings of Machine Learning and Systems, 4, 217-237.

---

> ### Author Response · Authors · 2023-11-16
> **Official answers to Reviewer 9edT**
>
> We thank reviewer 9edT for their helpful feedback. In the following, we address each of their concerns individually.
> ## Weaknesses:
> - We added an overview of our strategy in Algorithm 1 in the Appendix Section A.2 and improved the discussion on how the metric function is learned.
> - We conducted ablation studies on all relevant parameters: $T$, $\epsilon$, and $\lambda$ (refer to Section 4.1). It's crucial to note that T is intrinsic to the network architecture and, therefore, must not be considered a hyperparameter for Rel-U. Additionally, the introduction of additive noise $\epsilon$ serves the purpose of ensuring a fair comparison with Doctor/ODIN, where noise was utilized to enhance detection performance. Nevertheless, as indicated by the results in the ablation study illustrated in Figure 3, $\epsilon=0$ seems to be close to optimal most of the time, thereby positioning Rel-U as an effective black-box algorithm. Furthermore, Rel-U exhibits a considerable degree of insensitivity to various values of $\lambda$, as evident from Figure 3. This suggests that a potential selection for λ could have been $\lambda=N_+/(N_+ + N_-)$, aiming to balance the ratio between positive ($N_+$) and negative examples ($N_-$). In such a scenario, there are no hyper-parameters at all. This clarification has been incorporated into the revised version of the paper, cf. Appendix A.5.
>
> ## Questions:
> - We added an overview and algorithm block in the appendix Section A.2 and expanded discussion on assumptions and requirements.
> - Our metric doesn't directly incorporate uncertainty statistics obtained from the model, as relying on such statistics may create a misleading sense of the model's genuine, as opposed to self-assured, confidence in detecting its classification errors. Unlike metrics that depend on transformations applied to latent code, our approach restricts observation to black-box access of the model, where only the final layer statistics can be observed (with $\epsilon$ set to 0). Instead of adopting a strategy that involves learning a metric for uncertainty through an additional model (as seen in [1] for OOD), which might raise concerns about the uncertainty of this secondary model, our metric focuses on learning to recognize the signature of uncertainty for each pair of classes using the soft-prediction of the existent model. In essence, our observer '$d$', given a sufficient number of observed positive and negative samples, can effectively weigh the uncertainty metric based on the model's soft-probabilities in a manner specific to the class pair in question.
> - Experiments with a MLP trained on the output logits of the validation set to predict mistakes were performed. Please see Table 3 in the Appendix for details. We added additional details on the hyper-parameters and the exact architecture used in the new Section A.3 in the Appendix.
> - We compared to MCDropout [A] and Deep Ensembles [B], which are estimates of Bayesian Neural Networks, in Table 3 in the Appendix. We added further details on these baselines in Section A.3 in the appendix.
>
>    In contrast to [2], we are given a pre-trained model and we want to gauge the uncertainty w.r.t. pairs of labels. This is slightly different from measuring the uncertainty on a class of models or on a given label for a family of models. It is worth mentioning that our focus is to develop a methodology that can be applied to any classifier that outputs soft-predictions, which includes, but is not limited to (Bayesian) neural networks.
>
> We believe that all the concerns and questions were adequately addressed, and kindly ask the reviewer to consider raising their scores accordingly. We are happy to provide any further clarification where needed.
>
> **References:**
>
> [1] Dinari, O. & Freifeld, O.. (2022). Variational- and metric-based deep latent space for out-of-distribution detection. *Proceedings of the Thirty-Eighth Conference on Uncertainty in Artificial Intelligence*, in *Proceedings of Machine Learning Research* 180:569-578 Available from https://proceedings.mlr.press/v180/dinari22a.html.
>
> [2] Vadera, M., Li, J., Cobb, A., Jalaian, B., Abdelzaher, T., & Marlin, B. (2022). URSABench: A system for comprehensive benchmarking of Bayesian deep neural network models and inference methods. *Proceedings of Machine Learning and Systems*, 4, 217-237.
>
> [A] Gal, Yarin, and Zoubin Ghahramani. "Dropout as a Bayesian Approximation: Representing Model Uncertainty in Deep Learning." ICML 2016. /abs/1506.02142.
>
> [B] Lakshminarayanan, Balaji, Alexander Pritzel, and Charles Blundell. "Simple and Scalable Predictive Uncertainty Estimation Using Deep Ensembles." NIPS 2017. /abs/1612.01474.

---

> > ### Comment · Reviewer_9edT · 2023-11-22
> > **Response to the rebuttal.**
> >
> > Thank you for your response.
> >
> > I guess I don't fully understand as to how this can be called a black-box approach (when you still need the final layer statistics). Can you try responding to my second point in the questions differently?
> >
> > Also, I acknowledge the responses to my other questions - the response is sufficient to alleviate my concerns on them. Please summarize our discussion and additional references for BNNs from the review cycle and add it to the paper - it'll be helpful to contrast this work with the rest.

---

> ### Author Response · Authors · 2023-11-22
> **Re: Response to the rebuttal.**
>
> Thank you for carefully reviewing our answers and the revised manuscript. Next, we address the follow-up questions.
>
> - To appropriately contextualize our contribution in the existing literature, initially, we adhered to the glossary provided by [1], which defines black-box methods as those relying solely on the final layer statistics. To address this valid concern, we revised section A.5 by eliminating the term "black-box" and rephrasing the corresponding sentence.
> - Broadly, our paper aligns with the extensive literature that leverages the model's statistics to articulate the uncertainty of the model's decision (e.g., "logit/feature transformation on held-out set"). The novelty introduced by our method is fundamentally characterized by two main aspects:
>     - As pointed out by the reviewer, our approach suggests a closed-form solution, that distinguishes it from [2], where an iterative algorithm is necessary to train a generative model (VAE) for estimating the likelihood of a sample. Notably, our method alleviates the need for practitioners to specify hyperparameters such as learning rate, number of layers, number of nodes, and number of epochs.
>
>     - Instead of directly calculating an uncertainty score based on the model's statistics (see [1, 3]), our data-driven approach optimizes a positive definite matrix D for the purpose of identifying misclassifications. This optimization implicitly defines an inner product to gauge similarities between vectors of softmax outputs.
>
> ## Summary of discussion and added comment in the paper
>
> Following the suggestion of the reviewer, we summarized the content of our discussion and added to the manuscript (in blue) as follows:
> We added a reference to the algorithm that clarifies the description of the computation of D (see Section 4.2 and Section A.2).
> To summarize our discussion on the Bayesian NN (BNN), we added relative literature to the related works section, and we introduced the results for MC dropout, deep ensemble, and MLP in section 5.1, also referencing our extended results in Appendix A.6.
> We clarified the hyperparameter tuning procedure, and provided further details on the ablation studies in Appendix A.5.
>
>
> **References:**
>
> [1] Granese F., Romanelli M., Gorla D., Palamidessi C., and Piantanida P. “DOCTOR: A simple method for detecting misclassification errors”. NeurIPS, 2021 \
> [2] Dinari, O. & Freifeld, O. (2022). Variational- and metric-based deep latent space for out-of-distribution detection. *Proceedings of the Thirty-Eighth Conference on Uncertainty in Artificial Intelligence*, in *Proceedings of Machine Learning Research* 180:569-578 \
> [3] Kimin Lee, Kibok Lee, Honglak Lee, Jinwoo Shin. A Simple Unified Framework for Detecting Out-of-Distribution Samples and Adversarial Attacks. NIPS, 2018.

---

> > ### Author Response · Authors · 2023-11-23
> >
> > We hope that the new answer satisfactorily addresses the remaining concerns, and we kindly ask reviewer 9edT to update their score accordingly.

---

### Meta-Review · Area_Chair_2rFH · 2023-12-06

**Metareview:**

The paper introduces a data-driven measure of uncertainty relative to an observer for misclassification detection. By leveraging patterns learned in the distribution of soft-predictions through a post-training optimization process, the proposed uncertainty measure can identify misclassified samples based on the predicted class probabilities. Experiments conducted on multiple image classification
tasks demonstrate that the proposed method outperforms existing misclassification detection baselines.

The authors' rebuttal has addressed most concerns raised by the reviewers in their original reviews. All the reviewers participated in the discussions with the authors during the rebuttal phase, which helped to further clarify some confusions. The paper's quality has also been improved accordingly. At the end of author-reviewer discussion phase, all the reviewers were on the positive side, as reflected by their final ratings of the paper. The authors are encouraged to incorporate all the important discussion points to the final version of the paper.

**Justification For Why Not Higher Score:**

A more precise interpretation of the $d(y,y')$ term could help to better understand the proposed proposed relative uncertainty. Furthermore, the proposed approach relies on a decent-sized validation dataset (to solve the optimization problem) in order compute the matrix $D$ for relative uncertainty evaluation.

**Justification For Why Not Lower Score:**

The proposed relative uncertainty provides an interesting and new metric for misclassification detection.

---

### Decision · Program_Chairs · 2024-01-16

Accept (poster)